# Mitigating Reward Over-optimization in Direct Alignment Algorithms with Importance Sampling

**Phuc Minh Nguyen**[1]**, Ngoc-Hieu Nguyen**[1]**, Duy H. M. Nguyen**[3,7,9]**, Anji Liu**[3,4]**, An Mai**[5]**,**
**Binh T. Nguyen**[6]**, Daniel Sonntag**[7,8]**, Khoa D. Doan**[1,2]
[1]College of Engineering and Computer Science, VinUniversity
[2]VinUni-Illinois Smart Health Center, VinUniversity
[3]University of Stuttgart, [4]National University of Singapore
[5]International University - VNUHCM, [6]University of Science - VNUHCM
[7]German Research Center for Artificial Intelligence (DFKI), [8]Oldenburg University
[9]Max Planck Research School for Intelligent Systems (IMPRS-IS)

## Abstract

Direct Alignment Algorithms (DAAs) such as Direct Preference Optimization (DPO) have emerged as alternatives to the standard Reinforcement Learning from Human Feedback (RLHF) for aligning large language models (LLMs) with human values. However, these methods are more susceptible to over-optimization, in which the model drifts away from the reference policy, leading to degraded performance as training progresses. This paper proposes a novel importance-sampling approach to mitigate the over-optimization problem of offline DAAs. This approach, called (IS-DAAs), multiplies the DAA objective with an importance ratio that accounts for the reference policy distribution. IS-DAAs additionally avoid the high variance issue associated with importance sampling by clipping the importance ratio to a maximum value. Our extensive experiments demonstrate that IS-DAAs can effectively mitigate over-optimization, especially under low regularization strength, and achieve better performance than other methods designed to address this problem. Our code is available at `https://github.com/mail-research/AIS-Sampling4DAAs`.

## 1 Introduction

Preference learning has emerged as an important part of the fine-tuning process to align large language models (LLMs) with human preferences. There are two predominant flavors of preference learning for LLMs. The first approach is online reinforcement learning from human feedback (RLHF) [Ouyang et al., 2022, Christiano et al., 2017]. RLHF typically involves a multi-stage procedure: fine-tuning a reward model to capture human preference and fine-tuning the LM policy to maximize the expected reward using online RL algorithms such as Proximal Policy Optimization [Schulman et al., 2017]. While empirically exhibiting good performance, this multi-stage procedure is complex and computationally intensive, requiring repeated queries of the reward model and sampling from the current policy. A set of alternative methods called direct alignment algorithms (DAAs) [Rafailov et al., 2024, Tang et al., 2024c] avoids fitting separate reward models and instead simply trains the policy directly on the offline preference dataset. The most well-known examples are Direct Preference Optimization (DPO) [Rafailov et al., 2023], and Identity Preference Optimization (IPO) [Tang et al., 2024b]. Since DAAs typically do not sample new responses from the LLM's policy during training, they are characterized as offline preference learning.

These methods are often preferred when aligning LLMs thanks to their simple training pipeline. However, recent studies find that DAAs are more susceptible to over-optimization—a phenomenon in which performance degrades as training progresses—and underperform online methods [Rafailov et al., 2024, Park et al., 2024a, Liu et al., 2024b, Tang et al., 2024a]. More specifically, Rafailov

et al. [2024] and Tang et al. [2024a] use the KL divergence between the preference-trained policy and reference SFT policy as a measure of budget and show that, as this budget increases, DAAs tend to experience a greater decline in performance. Rafailov et al. [2024] explains over-optimization in DAAs by the lack of training data coverage (i.e., using only offline samples); thus, optimizing their loss function can push up the probability of responses that are out of the offline data distribution. This is not the case in online alignment algorithms, since they can sample online data and explicitly enforce reverse KL regularization to mitigate this behavior [Song et al., 2024].

In this work, we identify an important source of the performance gap between online alignment algorithms and DAAs: during training, the trained policy gradually diverges from its initial distribution used to collect preference data. Meanwhile, the regularization effect in DAAs provides only a local approximation of KL regularization around this offline distribution, which is insufficient to prevent this drift. One approach to mitigate this problem is to enforce an explicit KL divergence penalty to encourage the model to stay close to the reference policy [Song et al., 2024, Fisch et al., 2024, Ding et al., 2024] and train the policy in an online manner [Guo et al., 2024]. This regularization explicitly prevents the LM policy from deviating from the initial distribution. However, these methods are costly since they require repeated sampling from the current policy to estimate the KL divergence and are sensitive to hyperparameters. Instead, this paper proposes a novel importance-sampling approach, called Importance Sampling DAAs (IS-DAAs). Furthermore, the implementation of IS incurs minimal computational overhead, making it highly scalable.

**Our main contributions are:** First, we investigate the over-optimization problem in offline alignment methods and propose a novel importance sampling method to address this problem for DAAs. Then, we fix the high-variance problem associated with importance sampling by clipping the importance ratios. Finally, our extensive experimental results indicate that IS-DAAs outperform DAAs and other regularized methods. More importantly, IS-DAAs are significantly better in mitigating over-optimization and early convergence issues, compared to previous regularization approaches.

## 2 Preliminaries

We provide the formulation and background of RLHF and DAAs in sections 2.1 and 2.2, respectively. The over-optimization phenomenon and regularization in DAAs are presented in section 2.3.

### 2.1 Reinforcement Learning from Human Feedback

To align an LM with human preferences, the RLHF pipeline consists of three stages:

**Supervised Fine-Tuning (SFT)**: Given a pre-trained model and a dataset of prompts $\mathbf{x}$ and their responses $\mathbf{y}$, the LM is trained for instruction following using maximum likelihood estimation (MLE) over next-tokens, resulting in *the reference model* $\pi_{\text{ref}}(\mathbf{y}|\mathbf{x})$.

**Reward Modeling**: In the second phase, the reference model $\pi_{\text{ref}}$ is prompted with $\mathbf{x}$ to produce a pair of responses $(\mathbf{y_1}, \mathbf{y_2}) \sim \pi_{\text{ref}}(\cdot|\mathbf{x})$. These responses are then labeled by human experts; the resulting pair is denoted as $\mathbf{y}^w \succ \mathbf{y}^l|\mathbf{x}$, expressing human preference for $\mathbf{y}^w$ over $\mathbf{y}^l$. This collected preference data is denoted as $\mathcal{D} = \{\mathbf{x}^{(i)}, \mathbf{y}^{w(i)}, \mathbf{y}^{l(i)}\}_{i=1}^N$. Typically, preference labels are assumed to follow the Bradley-Terry model:

$$p(\mathbf{y}_1 \succ \mathbf{y}_2|x) = \frac{\exp(r(\mathbf{x}, \mathbf{y}_1))}{\exp(r(\mathbf{x}, \mathbf{y}_1)) + \exp(r(\mathbf{x}, \mathbf{y}_2))} = \sigma\big(r(\mathbf{x}, \mathbf{y}_1) - r(\mathbf{x}, \mathbf{y}_2)\big) \tag{1}$$

where $r(\mathbf{x}, \mathbf{y})$ is a reward model, mapping a pair of an input prompt and its response to a scalar reward. We can then use $\mathcal{D}$ to train a parametrized reward model $r_\phi(\mathbf{x}, \mathbf{y})$ to maximize the differences between the rewards of $\mathbf{y}^w$ and $\mathbf{y}^l$ using MLE, as follows:

$$\mathcal{L}_R(r_\phi) = -\mathbb{E}_{(\mathbf{x}, \mathbf{y}^w, \mathbf{y}^l) \sim \mathcal{D}}[\log \sigma\big(r_\phi(\mathbf{x}, \mathbf{y}^w) - r_\phi(\mathbf{x}, \mathbf{y}^l)\big)] \tag{2}$$

**RL Fine-tuning**: The learned reward function is used to provide feedback in the RL phase, using an on-policy algorithm, such as PPO, with the following objective:

$$\max_{\pi_\theta} \mathbb{E}_{\mathbf{x} \sim \mathcal{D}, \mathbf{y} \sim \pi_\theta(\cdot|\mathbf{x})} \Big[r_\phi(\mathbf{x}, \mathbf{y}) - \beta \mathbb{KL}(\pi_\theta || \pi_{\text{ref}})\Big] \tag{3}$$

where $\pi_\theta$ is the learning policy, and $\beta$ is the hyper-parameter controlling the KL regularization w.r.t the reference policy $\pi_{\text{ref}}$. This KL constraint prevents the model from deviating too far from the region on which the reward model is well-trained, and mode-collapsing to only high-reward responses.

## 2.2 Direct Alignment Algorithms (DAAs)

Despite its superior performance in aligning the LMs with human preferences, the RLHF pipeline is complex and computationally expensive. DAAs address these problems by directly optimizing the policy $\pi_\theta$ over the preference data, avoiding the reward function estimation and RL training phases. Among these algorithms, DPO is the most popular approach; DPO derives the following closed-form solution for $\pi^\star$ of Eqn. (3):

$$\pi^*(\mathbf{y}|\mathbf{x}) = \frac{1}{Z(\mathbf{x})} \pi_{\text{ref}}(\mathbf{y}|\mathbf{x}) \exp\left(\frac{1}{\beta} r(\mathbf{x}, \mathbf{y})\right) \tag{4}$$

where $Z(\mathbf{x})$ as the normalization function. According to the above equation, we can parameterize the reward function by the log-likelihood ratio between $\pi_\theta$ and $\pi_{\text{ref}}$:

$$r_\theta(\mathbf{x}, \mathbf{y}) = \beta \log \frac{\pi_\theta(\mathbf{y}|\mathbf{x})}{\pi_{\text{ref}}(\mathbf{y}|\mathbf{x})} + \beta \log Z(\mathbf{x}) \tag{5}$$

This enables us to optimize the LM policy $\pi_\theta$ directly on the preference data by plug the above equation into Eq. (1) and minimizing the corresponding negative-loglikelihood:

$$\mathcal{L}_{\text{DAA}}(\pi_\theta, \pi_{\text{ref}}) = \mathbb{E}_{(\mathbf{x}, \mathbf{y}^w, \mathbf{y}^l) \sim \mathcal{D}} \left[ f\left( \beta \log \frac{\pi_\theta(\mathbf{y}^w|\mathbf{x})}{\pi_{\text{ref}}(\mathbf{y}^w|\mathbf{x})} - \beta \log \frac{\pi_\theta(\mathbf{y}^l|\mathbf{x})}{\pi_{\text{ref}}(\mathbf{y}^l|\mathbf{x})} \right) \right] \tag{6}$$

where $f$ is a convex loss function. By choosing the loss function $f$, we can recover the standard DPO objective [Rafailov et al., 2023] ($f(x) = -\log\sigma(x)$), or the IPO objective [Azar et al., 2024] ($f(x) = (x-1)^2$). Other objectives can be found in [Tang et al., 2024b]. In this paper, we will focus on these two well-studied objectives.

## 2.3 Over-Optimization and the Performance Gap Between Online and Offline Alignments

In this study, we refer to online alignment methods as those that require sampling from the currently trained policy during the training process. This includes the two-stage RLHF and online variants of DPO or IPO [Guo et al., 2024, Schulman et al., 2017]. In contrast, offline methods do not require on-policy sampling and are computationally more efficient.

Experiments from Tang et al. [2024a] demonstrate that both online and offline alignment algorithms suffer from over-optimization, wherein an alignment algorithm consumes a large *optimization budget* (such as how far the optimized policy $\pi_\theta$ drifts away from the reference policy $\pi_{\text{ref}}$, measured by KL divergence) without improving (and even reducing) its performance. However, offline methods exhibit a larger degree of degradation compared to online methods. Importantly, they show that offline data coverage and data quality cannot convincingly explain the performance difference.

In this study, we hypothesize that this performance gap is caused by inappropriate regularization when using only off-policy data in offline DAAs, making these algorithms more susceptible to over-optimization. In the standard RLHF (Eqn. (3)), the trade-off between the surrogate reward scores and the negative KL divergence is explicitly formulated. In each optimization step, if the first term outweighs the second term, the learning algorithm optimizes the LM toward the direction that increases the surrogate reward scores at the cost of increasing the KL divergence budget. Consequently, if the surrogate reward function is not a good approximation of the true reward function, this behavior will result in an ineffective use of the KL-divergence budget, leading to over-optimization.

In DAAs, this tradeoff is implicit. To see this, we denote the log ratio difference in DAAs' loss function as $\rho_\theta := \log \frac{\pi_\theta(\mathbf{y}^w)}{\pi_{\text{ref}}(\mathbf{y}^w)} - \log \frac{\pi_\theta(\mathbf{y}^l)}{\pi_{\text{ref}}(\mathbf{y}^l)}$ and consider the Taylor expansion around $\rho_\theta = 0$, which is the case when the finetuning starts with $\pi_\theta = \pi_{\text{ref}}$,

$$\underbrace{\mathbb{E}_{(\mathbf{x}, \mathbf{y}^w, \mathbf{y}^l) \sim \mathcal{D}} \left[ f\left(\beta\rho_\theta\right) \right]}_{\text{direct alignment loss}} \approx f(0) + \underbrace{f'(0)\beta \cdot \mathbb{E}_{(\mathbf{x}, \mathbf{y}^w, \mathbf{y}^l) \sim \mathcal{D}} \left[\rho_\theta\right]}_{\text{preference optimization}} + \underbrace{\frac{f''(0)\beta^2}{2} \cdot \mathbb{E}_{(\mathbf{x}, \mathbf{y}^w, \mathbf{y}^l) \sim \mathcal{D}} \left[\rho_\theta^2\right]}_{\text{weighted squared loss}}, \tag{7}$$

Tang et al. [2024c] show that if $\mathbf{y}^w$ and $\mathbf{y}^l$ are generated by $\pi_\theta$ and a labeling function $p$, then the expectation of the gradient of the weighted squared loss term recovers the update of the reverse KL regularization, i.e.,

$$\mathbb{E}_{(\mathbf{x}, \mathbf{y}^w, \mathbf{y}^l) \sim (\mathcal{D}, \pi_\theta)} \left[ \nabla_\theta \frac{1}{2} \rho_\theta^2 \right] = C \cdot \mathbb{E}_{\mathbf{x} \sim \mathcal{D}} \left[ \nabla_\theta \mathbb{KL}\left( \pi_\theta(\cdot|\mathbf{x}), \pi_{\text{ref}}(\cdot|\mathbf{x}) \right) \right] \tag{8}$$

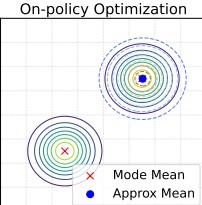 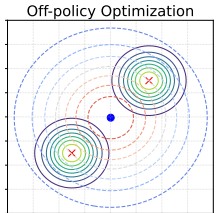 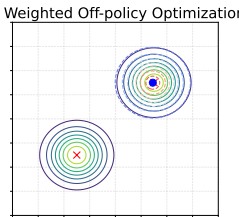

| On-policy Optimization | Off-policy Optimization | IS Weighted Off-policy Optimization |
|---|---|---|

Figure 1: The results of minimizing the weighted square loss term with on-policy data (left), off-policy data (middle), and off-policy data with importance sampling weights (right). The reference distribution is modeled by a mixture of two-dimensional Gaussians, and $\pi_\theta$ is an unimodal Gaussian.

where $C$ is a constant. This equality suggests that DAAs with *online preference data* can enforce regularization by optimizing the weighted squared loss. However, in the offline setting, i.e., $\mathbf{y}^w$ and $\mathbf{y}^l$ are generated by $\pi_{\text{ref}}$ and as $\pi_\theta$ moves away from $\pi_{\text{ref}}$, Eqn. (8) does not hold. Instead, we have

$$\mathbb{E}_{(\mathbf{x},\mathbf{y}^w,\mathbf{y}^l)\sim(\mathcal{D},\pi_{\text{ref}})}\left[\nabla_\theta \frac{1}{2}\rho_\theta^2\right]$$
$$=2\mathbb{E}_{(\mathbf{x},\mathbf{y})\sim(D,\pi_{\text{ref}})}\left[\log\frac{\pi_\theta(\mathbf{y}|\mathbf{x})}{\pi_{\text{ref}}(\mathbf{y}|\mathbf{x})}\nabla\log\pi_\theta(\mathbf{y}|\mathbf{x})\right]+$$
$$2D_{KL}\left[\pi_{\text{ref}}(\cdot|\mathbf{x})|\pi_\theta(\cdot|\mathbf{x})\right]\mathbb{E}_{(\mathbf{x},\mathbf{y})\sim(D,\pi_{\text{ref}})}\left[\nabla\log\pi_\theta(\mathbf{y}|\mathbf{x})\right] \quad (9)$$

The detailed derivation can be found in Appendix B.2. In Equation (9), since the KL divergence is always positive, going in the opposite direction of the second term will push down the probability of both winning and losing responses from the training data. Moreover, the first term will only regularize samples from $\pi_{\text{ref}}$. Consequently, optimizing the weighted squared loss using offline data increases the weights of responses that are out of the reference policy distribution, and this effect is stronger when $\pi_\theta$ deviates from $\pi_{\text{ref}}$. The visualization of this effect is plotted in Figure 1. Since preference labels for OOD responses are not available, increasing their probability is inappropriate.

## 3 Mitigating Over-optimization in DAAs with Importance Sampling

The regularization effect in DAAs is effective only when the policy $\pi_\theta$ remains close to the reference policy $\pi_{\text{ref}}$. Once $\pi_\theta$ moves away from $\pi_{\text{ref}}$, the influence of the regularization diminishes (Section 2.3) and this can be detrimental to performance. To mitigate this problem, a simple approach is to apply online sampling training to collect responses from the current policy $\pi_\theta$ [Guo et al., 2024].

$$\mathcal{L}_{\text{Online-DPO}}(\pi_\theta, \pi_{\text{ref}}) =$$
$$-\mathbb{E}_{\mathbf{x}\sim\mathcal{D},(\mathbf{y}^w,\mathbf{y}^l)\sim\pi_\theta(\cdot|\mathbf{x})}\left[\log\sigma\left(\beta\log\frac{\pi_\theta(\mathbf{y}^w|\mathbf{x})}{\pi_{\text{ref}}(\mathbf{y}^w|\mathbf{x})}-\beta\log\frac{\pi_\theta(\mathbf{y}^l|\mathbf{x})}{\pi_{\text{ref}}(\mathbf{y}^l|\mathbf{x})}\right)\right] \quad (10)$$

However, online training is considerably more complex than offline methods, as it involves training an explicit reward model and sampling from the LM policy that is being trained, incurring significant computational costs. [Rafailov et al., 2023]

### 3.1 Importance Sampling DAAs (IS-DAAs)

Motivated by the insight in Section 2.3, which reveals the inappropriate implicit KL regularization in offline DAAs, we design a method to more effectively estimate this KL divergence without requiring repeated online sampling from the $\pi_\theta$. Specifically, we leverage importance sampling to estimate the expectation under the LM policy $\pi_\theta$ distribution given samples from the offline data. This leads to the following objective:

$$\mathcal{L}_{\text{IS-DPO}}(\pi_\theta, \pi_{\text{ref}})$$
$$=-\mathbb{E}_{\mathbf{x}\sim\mathcal{D},\mathbf{y}^w,\mathbf{y}^l\sim\pi_{\text{ref}}(\cdot|\mathbf{x})}\left[w(\mathbf{x},\mathbf{y}^w,\mathbf{y}^l)\log\sigma\left(\beta\log\frac{\pi_\theta(\mathbf{y}^w|\mathbf{x})}{\pi_{\text{ref}}(\mathbf{y}^w|\mathbf{x})}-\beta\log\frac{\pi_\theta(\mathbf{y}^l|\mathbf{x})}{\pi_{\text{ref}}(\mathbf{y}^l|\mathbf{x})}\right)\right] \quad (11)$$

where the importance weights $w(\mathbf{x}, \mathbf{y}^w, \mathbf{y}^l) = \frac{\pi_\theta(\mathbf{y}^w|\mathbf{x})}{\pi_{\text{ref}}(\mathbf{y}^w|\mathbf{x})} \frac{\pi_\theta(\mathbf{y}^l|\mathbf{x})}{\pi_{\text{ref}}(\mathbf{y}^l|\mathbf{x})}$. Here, the importance weight is the ratio of sequence-level probability between $\pi_\theta$ and $\pi_{\text{ref}}$, i.e., $\frac{\pi_\theta(\mathbf{y}|\mathbf{x})}{\pi_{\text{ref}}(\mathbf{y}|\mathbf{x})} = \prod_{t=1}^{T} \frac{\pi_\theta(y_t|\mathbf{x},\mathbf{y}_{<t})}{\pi_{\text{ref}}(y_t|\mathbf{x},\mathbf{y}_{<t})}$. The update is multiplied by this importance weight to adjust the action probabilities so that the expectation according to the LM policy $\pi_\theta$ can be calculated with samples from $\pi_{\text{ref}}$.

We next show that by multiplying with an importance ratio, the objective in Eqn. (11) is an unbiased estimation of the objective in Eqn. (16) without requiring recursively sample from the learned policy. Moreover, this also leads to the gradient of weighted squared loss, and the gradient of KL divergence coincides under certain conditions. More formally, we have:

**Theorem 1** *Assuming Supp $(\pi_{ref})$ = Supp $(\pi_\theta)$, then the objective in Eqn.* (11) *is an unbiased estimation of Eqn.* (16) *and the gradient concerning $\pi_\theta$ of the weighted squared loss equals to the gradient of KL divergence,*

$$\mathbb{E}_{\mathbf{x}\sim\mathcal{D},(\mathbf{y}^w\mathbf{y}^l)\sim\pi_{ref}(\cdot|x)} \left[ w(\mathbf{x}, \mathbf{y}^w, \mathbf{y}^l)\nabla_\theta \frac{1}{2}\rho_\theta^2 \right] = \mathbb{E}_{\mathbf{x}\sim\mathcal{D}} \left[ \nabla_\theta \mathbb{KL} \left( \pi_\theta(\cdot|\mathbf{x}), \pi_{\text{ref}}(\cdot|\mathbf{x}) \right) \right] \quad (12)$$

Theorem 1 implies that by incorporating the importance ratio, we can ensure that minimizing the $\mu$-squared regularization also minimizes the KL divergence. The proof is in Appendix B.1.

**Mitigating Importance Sampling's high variance using clipping.** Directly computing the importance weights during training can lead to extremely high variance Elvira and Martino [2021], potentially resulting in gradient explosions due to extreme values. To mitigate this problem, we propose to use *Truncated importance weighting* estimator [Elvira and Martino, 2021], as follows:

$$w(\mathbf{x}, \mathbf{y}^w, \mathbf{y}^l) = \max \left( \frac{\pi_\theta(\mathbf{y}^w|\mathbf{x})}{\pi_{\text{ref}}(\mathbf{y}^w|\mathbf{x})} \frac{\pi_\theta(\mathbf{y}^l|\mathbf{x})}{\pi_{\text{ref}}(\mathbf{y}^l|\mathbf{x})}, \epsilon \right) \quad (13)$$

where $\epsilon$ serves as a regularization to trade-off between the bias and variance of the importance ratio.

## 3.2 An analysis of regularization effect in DAAs with Importance Sampling

This section provides a detailed example in which the weighted squared regularization term only serves as a local approximation of the KL divergence when $\pi_\theta$ is near $\pi_{\text{ref}}$ and, by incorporating an importance ratio, we can enforce a more effective regularization in DAAs.

We consider a multi-arm bandit problem with 4-action space $\mathcal{A} = \{a_0, a_1, a_2, a_3\}$, the reference model $\pi_{\text{ref}}$ as $\pi_{\text{ref}}(a_0) = 0.5, \pi_{\text{ref}}(a_1) = \pi_{\text{ref}}(a_2) = 0.1$, and $\pi_{\text{ref}}(a_3) = 0.3$ , and the reward as $r(a_0) = 0.5, r(a_1) = 1, r(a_j) = 0, \forall j \in \{2, 3\}$. We can obtain the optimal policy analytically as $\pi^*(a) \propto \pi_{\text{ref}}(a) \exp(1/\beta \times r(a))$. We then learn a parametrized policy $\pi_\theta(a)$ to imitate the optimal policy $\pi^*(a)$ by minimizing the KL divergence. KL $(\pi_\theta||\pi^*)$.

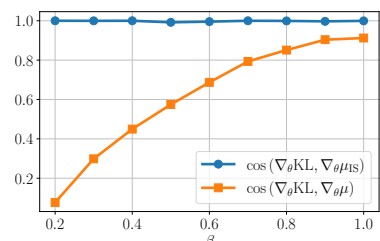

Figure 2: Cosine similarity $\cos(\nabla_\theta\text{KL}, \nabla_\theta\mu)$ and $\cos(\nabla_\theta\text{KL}, \nabla_\theta\mu_{\text{IS}})$ as a function of $\beta$.

To demonstrate that incorporating importance sampling can enforce KL regularization without the need for online sampling, we compare the gradients in terms of their cosine similarities. Specifically, we measure the cosine similarity between the gradient of the KL divergence with respect to the parameters of $\pi_\theta$ (denoted as $\nabla_\theta\text{KL}$) and the gradient of the weighted squared loss (denoted as $\nabla_\theta\mu$). This is expressed as $\cos(\nabla_\theta\text{KL}, \nabla_\theta\mu)$. Additionally, we compute the cosine similarity between the KL gradient and the gradient of the weighted squared loss with importance sampling (denoted as $\nabla_\theta\mu_{\text{IS}}$), represented as $\cos(\nabla_\theta\text{KL}, \nabla_\theta\mu_{\text{IS}})$. This analysis helps us understand how closely importance sampling aligns with the KL regularization as the learned policy $\pi_\theta$ moves away from $\pi_{\text{ref}}$. We experiment with $\beta \in [0.05, 1.0]$ to control the deviation of $\pi_\theta$ from $\pi_{\text{ref}}$.

As shown in Fig. 2, under high regularization, both the gradient $\nabla_\theta\mu_{\text{IS}}(\pi_\theta, \pi_{\text{ref}})$ and $\nabla_\theta\mu(\pi_\theta, \pi_{\text{ref}})$ exhibits high cosine similarity with $\nabla_\theta\text{KL}(\pi_\theta||\pi_{\text{ref}})$. However, the cosine similarity $\cos(\nabla_\theta\text{KL}, \nabla_\theta\mu)$ starts to decrease quickly when $\beta$ becomes too small and can even be negative. On the other hand, $\cos(\nabla_\theta\text{KL}, \nabla\mu_{\text{IS}})$ shows consistently high similarity across all $\beta$, this shows that by incorporating the importance ratio, we can enforce an effective regularization even when the learned policy deviate significantly from $\pi_{\text{ref}}$.

# 4 Experiments and Results

In this section, we empirically evaluate IS-DAAs ability to align language models with human preferences and mitigate the reward over-optimization problem. First, in the **TL;DR Summarization** task, we systematically study the trade-off between the policy performance and KL regularization achieved by different alignment methods in a controlled environment where we assume to have access to a golden reward model as the ground-truth preferences. Next, in the **Instruction Following** benchmark, we evaluate IS-DAAs on three standard open-ended instruction following benchmarks. Under both settings, IS-DAAs outperform existing alignment approaches and better mitigate the over-optimization problem compared to existing approaches designed for this purpose.

**Models**: Throughout our experiments, we use Llama-3.2-3B [MetaAI, 2024a,b] as the pre-trained base model. For both summarization and the instruction following, we first supervised fine-tuning Llama-3.2-3B to serve as the initialization for subsequent preference training.

**Baselines**: In addition to IS-DAAs, we evaluate several existing baselines that address the over-optimization problem in DAAs, including: DAAs+SFT objective [Liu et al., 2024a, Cen et al., 2025, Fisch et al., 2025], which augments the DAAs with an additional SFT loss, we refer to this approach as Regularized Preference Optimization (RPO), $\chi$-PO [Huang et al., 2025], which combines $\chi^2$ with KL regularization to enforce stronger regularization. Additionally, We also consider DAAs with length-regularization approach [Park et al., 2024a], to address the length exploitation issue – a common pattern of reward over-optimization [Park et al., 2024a, Chen et al., 2024b].

## 4.1 TL;DR Summarization

**Setup**: For the summarization task, we consider the filtered version of Reddit TL;DR summarization dataset [Stiennon et al., 2020]. Following the "controlled" environment setups of [Gao et al., 2022, Tang et al., 2024a, Rafailov et al., 2023], we assume access to the golden (ground-truth) reward model. This golden reward model is learned to encourage high-quality summaries of Reddit Posts. The golden reward model will provide preference feedback to create a synthetic preference dataset $\mathcal{D}_{\text{golden}} = \{\mathbf{x}_i, \mathbf{y}_i^w, \mathbf{y}_i^l\}_{i=1}^N$ for preference training.

The reference model is obtained through SFT on the dedicated SFT split. For preference training, we consider 2 epochs of training with varying regularization strength $\beta$ to investigate the reward maximization and KL divergence trade-off under different approaches.

**Evaluation**: We evaluate the performance of the learned policy by the win rate against the chosen summaries from $\mathcal{D}_{\text{golden}}$. The win-rate is determined by the golden reward model. Our main results are shown in Fig. 3, which presents the win rate and KL trade-off across different configurations after 2 epochs of training. Our main findings are:

**Previous approaches is insufficient to mitigate Reward Over-optimization.** As shown in Fig. 3, under low KL constraint strength ($\beta = 0.01$), apart from $\chi$-po, prior alignment methods – even with their proposed regularization techniques – still suffer from the over-optimization problem. These methods fail to constrain the learned policy to stay close to $\pi_{\text{ref}}$, leading to decreasing performance of the learned policy. Moreover, the results in Fig. 4 demonstrate the early convergence phenomenon [Park et al., 2024a], where these methods reach their peak performance after training on only $30-40\%$ of the data, followed by performance degradation as the training progresses with increasing KL divergence.

**IS-DPO is significantly more KL efficient than other methods.** Considering the square root KL divergence as the resource to be spent, we can observe that IS-DPO "spent" KL more effectively than the other methods. Under low regularization, IS-DPO effectively regularizes the policy to a significantly low KL budget and avoid reward over-optimization. Under stronger KL regularization ($\beta = 0.1$), IS-DPO spends a higher KL budget compared to the other approaches to achieve a better performance (Fig. 4). Interestingly, the effect of the penalty in IS-DPO is akin to the early stopping phenomenon (Fig. 3), where the training stops at a specific KL divergence $\sqrt{\text{KL}\left(\pi_\theta || \pi_{\text{ref}}\right)} (\approx 5.5)$ to avoid over-optimization without requiring any evaluation set. We discuss this result further in Section 4.3 of the Appendix.

**IS-DPO exhibits better performance while effectively mitigating early convergence problem.** As shown in Fig. 4, across different regularization strengths, IS-DPO achieves the highest averaged win rates. Furthermore, while other approaches suffer from an early convergence problem, IS-DPO

continues to improve or stay flat at later epochs with a significantly lower KL budget. This shows that IS-DPO is robust to the over-optimization problem.

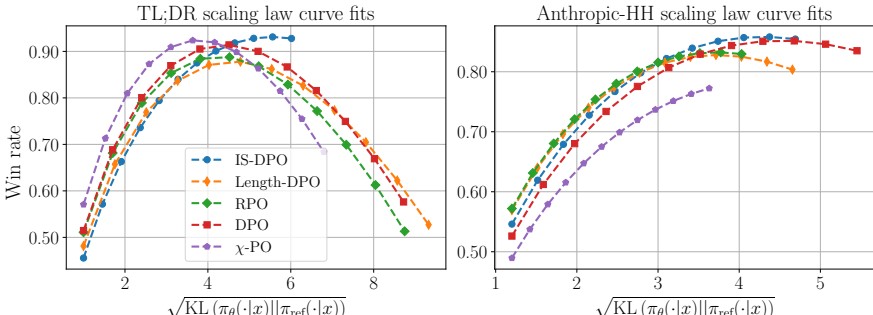

Figure 3: Result on over-optimization of different methods in TL;DR and Anthropic datasets. Results show the win rate over the reference summary as judged by the golden reward model as a function of square root KL divergence, with the proposed fitted curve from gao2022scalinglawsrewardmodel.

**Clipping ratio $\epsilon$ Ablation.** We conduct ablation studies to better understand how the clipping ratio $\epsilon$ influences policy performance and regularization in Fig. 5. As can be observed, a moderate value of $\log \epsilon = 1.0$ yields the highest win rate ($\approx 92\%$) and is associated with the highest KL divergence. These results are consistent with the bias-variance trade-off of clipping ratio $\epsilon$. Specifically, a higher value of $\epsilon$ allows a larger model update and better reflects the on-policy objective. However, setting $\epsilon$ too large can lead to only a small number of samples with excessively large weights that dominate the learning signals of the other valuable samples [Park et al., 2024b].

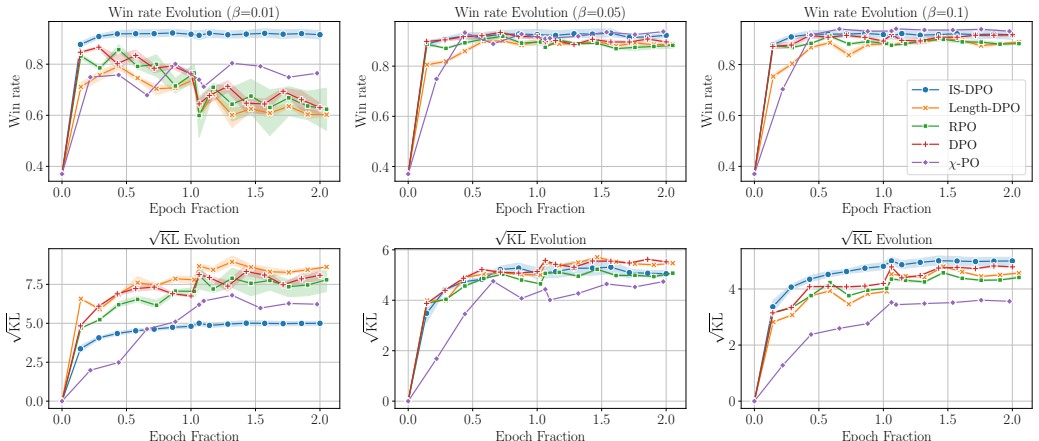

Figure 4: Results on optimization dynamics of different methods. The top row shows win rate over 2 epochs, while the bottom row shows the corresponding square root KL divergence. The shaded area displays standard error over 3 seeds. Under low KL regularization strength, IS-DPO can achieve better performance across different baselines and exhibit no over-optimization phenomenon while maintaining a significantly lower KL budget.

## 4.2 Instruction Following

**Setup**: For the instruction following task, we consider the Anthropic Helpful and Harmless (HH) dataset [Bai et al., 2022], UltraFeedback dataset [] for preference trainining. We consider 1 epoch of training with $\beta = 0.05$ as standard configurations for offline alignments [Rafailov et al., 2024, Gao et al., 2024, Guo et al., 2024].

**Evaluation**: We evaluate methods fine-tuned on the UltraFeedback on 2 widely-adopted benchmarks: AlpacaEval 2.0 [Dubois et al., 2025] and MT-Bench [Zheng et al., 2023]. For Anthropic-HH, We evaluate the performance using a reward model trained on a preference dataset and querying GPT-4 to serve as a proxy for human evaluation. We provide a detailed evaluation setup in Section D of the Appendix. Similar to the summarization task, we compare the learned policy against the chosen responses from the preference data and report the win rate of various methods using GPT-4 and the trained reward model.

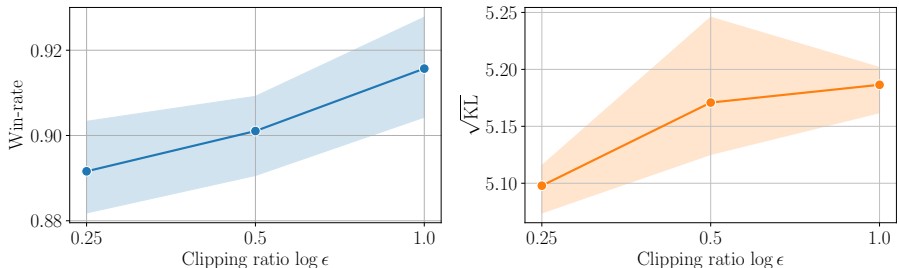

Figure 5: The ablation results for the clipping ratio $\epsilon$ in Eq (13). Results show model win rate and KL regularization effect over the summarization dataset across 3 random seeds. We observe that higher clipping ratios allow for larger policy updates, resulting in both increased KL divergence between the policy and reference model and improved win rate.

**Results**: Experimental results are shown in Tab. 1. For Anthropic-HH, IS-DPO demonstrates notable improvements over the other baselines, in both evaluations using both the reward model and GPT-4. Specifically, IS-DPO achieves approximately 3% improvement over the standard DPO, judged by GPT-4 and the reward model. On AlpacaEval 2.0 and MT-Bench, our results show that IS-DPO consistently outperforms state-of-the-art methods such as DPO and RPO, and demonstrate competitive results and achieve competitive performance with $\chi$-PO on AlpacaEval 2.0.

| Method | Anthropic-HH | | AlpacaEval 2.0 (%) | MT-Bench |
| | RM (%) | GPT-4 WR (%) | GPT-4 WR (%) | GPT-4 Score |
|---|---|---|---|---|
| DPO | $81.77 \pm 0.5$ | $71.1 \pm 0.6$ | 5.81 | 5.39 |
| RPO | $81.50 \pm 0.4$ | $66.5 \pm 0.4$ | 5.77 | 5.29 |
| Length-DPO | $83.85 \pm 0.8$ | $58.0 \pm 1.0$ | 3.71 | 5.24 |
| $\chi$-PO | $81.25 \pm 1.5$ | $67.2 \pm 1.4$ | **6.85** | 5.09 |
| IS-DPO (Ours) | $\mathbf{84.60 \pm 0.6}$ | $\mathbf{74.0 \pm 0.9}$ | 6.33 | **5.44** |

Table 1: Average win rate and standard deviation from 3 different seeds against the chosen responses on the different dialogue datasets. The best results are **bolded**.

## 4.3   Importance Sampling as Support Constraint

This section analyzes why Importance Sampling can serve as an implicit early stopping mechanism, effectively avoiding over-optimization. Consider 2 discrete distributions $P(X)$ and $Q(X)$ defined over a random variable $X$. We wish to estimate a function $f(x)$ under $P$: $\mathbb{E}_{x \sim P}[f(x)]$ and IS achieves this estimation through $\mathbb{E}_{x \sim Q}\left[\frac{P(x)}{Q(x)}f(x)\right]$. Under the support assumption $\text{supp}(P) \subseteq \text{supp}(Q)$, IS will be an unbiased estimation. However, when this assumption does not hold, IS instead estimates $\sum_{\text{supp}(Q)} P(x)f(x)$, which is biased.

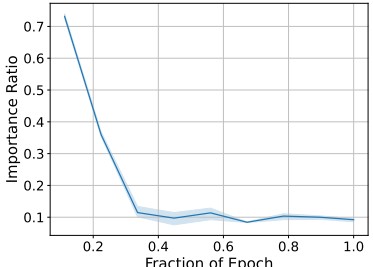

Figure 6: Averaged importance ratio $\pi_\theta(\mathbf{y}|\mathbf{x})/\pi_{\text{ref}}(\mathbf{y}|\mathbf{x})$ during training.

In the case of IS-DAAs, where $\pi_\theta$ is $P$ and $\pi_{\text{ref}}$ is $Q$, this biased estimate can be a blessing in disguise. To elaborate, minimizing the objective in Eqn. (11) is equivalent to minimizing:

$$\mathbb{E}_{\mathbf{x} \sim \mathcal{D}, \mathbf{y}^w, \mathbf{y}^l \sim \tilde{\pi}_\theta(\cdot|\mathbf{x})}\left[\log \sigma\left(\beta \log \frac{\pi_\theta(\mathbf{y}^w|\mathbf{x})}{\pi_{\text{ref}}(\mathbf{y}^w|\mathbf{x})} - \beta \log \frac{\pi_\theta(\mathbf{y}^l|\mathbf{x})}{\pi_{\text{ref}}(\mathbf{y}^l|\mathbf{x})}\right)\right] \qquad (14)$$

where $\tilde{\pi}$ is defined as:

$$\tilde{\pi}_\theta(\mathbf{y}|\mathbf{x}) = \frac{\mathbb{1}\left[\pi_{\text{ref}}(\mathbf{y}|\mathbf{x}) > 0\right]\pi_\theta(\mathbf{y}|\mathbf{x})}{\sum_{\mathbf{y}} \mathbb{1}\left[\pi_{\text{ref}}(\mathbf{y}|\mathbf{x}) > 0\right]\pi_\theta(\mathbf{y}|\mathbf{x})} \qquad (15)$$

By adopting Importance sampling, IS-DAAs only provide updates to $\pi_\theta$ when $\pi_\theta$'s samples are in $\pi_{\text{ref}}$'s support regions. When the support assumption is violated, $\text{supp}(\pi_\theta) \not\subseteq \text{supp}(\pi_{\text{ref}})$, IS-DAAs will not update $\pi_\theta$, thus avoiding the extrapolation issue.

Empirically, we observe that, throughout the training process, the importance ratio $\pi_\theta/\pi_{\mathrm{ref}}$ decreases as training progresses (Fig. 6). This means that several samples have low probability under $\pi_\theta$, or $\pi_\theta(\mathbf{y}|\mathbf{x}) \approx 0$. This leads to $\pi_\theta$'s updates with fewer and fewer samples, inducing an early stopping effect. Without IS, further training on these samples with small importance ratios ($\pi_\theta/\pi_{\mathrm{ref}}$) can lead to performance degradation.

## 5 Related Works

### 5.1 Reinforcement Learning from Human Feedback

In recent years, RLHF has been a dominant framework for aligning LLMs with human preferences. The RLHF pipeline begins with supervised fine-tuning (SFT) of the LLM using next-token prediction objectives on a dataset of high-quality, instruction-following responses. This is followed by fine-tuning the SFT's LLM using reinforcement learning (RL) algorithms such as PPO [Schulman et al., 2017] or REINFORCE [Williams, 2004], to maximize an "explicit reward" (based on the preference data) with a KL regularization to the reference policy. Alternatively, DAAs, such as DPO [Rafailov et al., 2023] and IPO [Azar et al., 2024], aim to simplify the RLHF pipeline by directly optimizing the LLM on human preferences without an explicit reward model or RL.

### 5.2 Over-optimization in DAAs

Gao et al. [2022] refer to *over-optimization* as the situation where an algorithm consumes a large *optimization budget* without improving (and even reducing) its performance. In this study, the KL divergence $\mathrm{KL}\left(\pi_\theta || \pi_{\mathrm{ref}}\right)$ is used as an optimization budget since it measures how far the optimized policy $\pi_\theta$ drifts away from the reference policy $\pi_{\mathrm{ref}}$ during training. Rafailov et al. [2024] study the trade-off between the KL divergence and the policy performance under three DAA objectives: DPO, IPO, and SLiC. They observe clear over-optimization after a certain point during training when an additional increase in the KL budget leads to a decrease in the model performance. This pattern persists across model sizes, with smaller models often exhibiting clearer signs of over-optimization. Regularization methods, such as length regularization park2024disentangling, rafailov2024scaling, can not mitigate this problem. Furthermore, Tang et al. [2024a] observes that both online and offline variants of DAAs suffer from over-optimization, while the online DAAs achieve better budget and performance trade-offs than the offline kinds.

### 5.3 Performance gap between online and offline alignment

We connect the over-optimization problem in offline alignment algorithms to the distribution shift problem encountered in offline RL [Levine et al., 2020, Kumar et al., 2020]. Specifically, during training, the policy $\pi_\theta$ is optimized using data generated by the reference model $\pi_{\mathrm{ref}}$. However, during deployment, the policy must act based on its own generated distribution, which may differ significantly from the training data. This discrepancy can lead to performance degradation, especially when the LMs encounter states that differ significantly from those in the offline data [Chen et al., 2024a]. While (DAAs) are designed for off-policy learning, they still suffer from this distribution shift problem [Rafailov et al., 2024, Tang et al., 2024c]. Our work demonstrates that, under such conditions, DAAs exhibit weak regularization and fail to regularize the learned policy when the LM deviates far away from its original (reference) distribution. Inspired by this observation, we propose IS-DAAs, which estimate an *on-policy* learning objective given only the *offline* preference data and effectively mitigate the distribution shift problem in DAAs.

## 6 Discussion

We study reward over-optimization in Direct Alignment Algorithms (DAAs). We show that one of the main sources of reward over-optimization in DAAs is the mismatch between offline distribution and the LM policy. To reduce this distribution gap problem, we introduce IS-DAAs, a simple yet effective method to estimate samples under the LM policy distribution given samples from the offline distribution. The proposed method is also able to overcome the high variance issue of importance ratio estimation. Our results showed that IS-DAAs outperform other regularization methods and effectively resolve the over-optimization issues in DAAs.

## Acknowledgements

This project is supported by VinUni-Illinois Smart Health Center (VISHC), VinUniversity. Duy M. H. Nguyen and Daniel Sonntag are supported by the No-IDLE project (BMBF, 01IW23002), the MASTER project (EU, 101093079), and the Endowed Chair of Applied Artificial Intelligence, Oldenburg University. Anji Liu is supported by the National University of Singapore under its Start-up Grant (Award No: SUG-251RES2505).

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

## A    Limitations and Societal Impacts

Our discussion highlights an ineffective regularization with direct alignment algorithms used widely to align to human preferences. In this work we analyze and resolve these issues. However, We still assume an underlying Bradley-Terry model of human preferences, as these models might not be accurate in explaining the ways humans give preferences and do not experiment with larger models due to limited computational resources. Furthermore, our IS-DPO implicitly assume hat the policy produced the offline dataset $\mathcal{D}$ is generated from $\pi_{\text{ref}}$, while might not holds in practice. Our work is to advance alignment algorithms that avoid over-optimization and ensure the development of models that are safe for real-world deployment.

## B    Proofs and Derivations

### B.1    IS-DPO is an unbiased estimation of online DPO

**Theorem 1 Restated.** *Assuming Supp $(\pi_{ref})$ = Supp $(\pi_\theta)$, then objective (Eqn. (11)) is an unbiased estimation of the objective in Eqn. (16) and the gradient concerning $\pi_\theta$ of the weighted squared loss equals to the gradient of KL divergence. Proof.* Online-DPO objective can be expressed as:

$$\mathcal{L}_{\text{Online-DPO}}(\pi_\theta, \pi_{\text{ref}}) =$$
$$- \mathbb{E}_{\mathbf{x}\sim\mathcal{D},(\mathbf{y}^w,\mathbf{y}^l)\sim\pi_\theta(\cdot|\mathbf{x})} \left[ \log \sigma \left( \beta \log \frac{\pi_\theta(\mathbf{y}^w|\mathbf{x})}{\pi_{\text{ref}}(\mathbf{y}^w|\mathbf{x})} - \beta \log \frac{\pi_\theta(\mathbf{y}^l|\mathbf{x})}{\pi_{\text{ref}}(\mathbf{y}^l|\mathbf{x})} \right) \right]$$

Expanding the expectation leads:

$$= -\mathbb{E}_{\mathbf{x}\sim\mathcal{D}} \left[ \sum_{\mathbf{y}^w,\mathbf{y}^l} \pi_\theta(\mathbf{y}^w,\mathbf{y}^l|\mathbf{x}) \log \sigma \left( \beta \log \frac{\pi_\theta(\mathbf{y}^w|\mathbf{x})}{\pi_{\text{ref}}(\mathbf{y}^w|\mathbf{x})} - \beta \log \frac{\pi_\theta(\mathbf{y}^l|\mathbf{x})}{\pi_{\text{ref}}(\mathbf{y}^l|\mathbf{x})} \right) \right]$$

$$= \mathbb{E}_{\mathbf{x}\sim\mathcal{D}} \left[ \sum_{\mathbf{y}^w,\mathbf{y}^l} \pi_{\text{ref}}(\mathbf{y}^w,\mathbf{y}^l|\mathbf{x}) \frac{\pi_\theta(\mathbf{y}^w,\mathbf{y}^l|\mathbf{x})}{\pi_{\text{ref}}(\mathbf{y}^w,\mathbf{y}^l|\mathbf{x})} \log \sigma \left( \beta \log \frac{\pi_\theta(\mathbf{y}^w|\mathbf{x})}{\pi_{\text{ref}}(\mathbf{y}^w|\mathbf{x})} - \beta \log \frac{\pi_\theta(\mathbf{y}^l|\mathbf{x})}{\pi_{\text{ref}}(\mathbf{y}^l|\mathbf{x})} \right) \right]$$

$$= \mathbb{E}_{\mathbf{x}\sim\mathcal{D}} \left[ \mathbb{E}_{(\mathbf{y}^w,\mathbf{y}^l)\sim\pi_{\text{ref}}(\cdot|\mathbf{x})} \left[ \frac{\pi_\theta(\mathbf{y}^w,\mathbf{y}^l|\mathbf{x})}{\pi_{\text{ref}}(\mathbf{y}^w,\mathbf{y}^l|\mathbf{x})} \log \sigma \left( \beta \log \frac{\pi_\theta(\mathbf{y}^w|\mathbf{x})}{\pi_{\text{ref}}(\mathbf{y}^w|\mathbf{x})} - \beta \log \frac{\pi_\theta(\mathbf{y}^l|\mathbf{x})}{\pi_{\text{ref}}(\mathbf{y}^l|\mathbf{x})} \right) \right] \right]$$

Where we denote $\pi(\mathbf{y}^w, \mathbf{y}^l) = \pi(\mathbf{y}^w|\mathbf{x})\pi(\mathbf{y}^l|\mathbf{x})$. This yields Eqn. (11).

Similarly, given a prompt $\mathbf{x}$, consider the gradient of KL divergence:

$$\nabla_\theta \text{KL}(\pi_\theta||\pi_{\text{ref}}) = \sum_{\mathbf{y}} \nabla_\theta \pi_\theta(\mathbf{y}|\mathbf{x}) + \sum_{\mathbf{y}} \log \left( \frac{\pi_\theta(\mathbf{y}|\mathbf{x})}{\pi_{\text{ref}}(\mathbf{y}|\mathbf{x})} \right) \nabla_\theta \pi_\theta(\mathbf{y}|\mathbf{x})$$

We can drop the first term since $\sum_{\mathbf{y}} \nabla_\theta \pi_\theta(\mathbf{y}|\mathbf{x} = \nabla_\theta \left( \sum_{\mathbf{y}} \pi_\theta(\mathbf{y}|\mathbf{x}) \right) = \nabla_\theta(1) = 0$. We now consider the gradient of weighted-squared loss with Importance Sampling:

$$\frac{1}{2}\mathbb{E}_{\mathbf{x}\sim\mathcal{D},(\mathbf{y}^w,\mathbf{y}^l)\sim\pi_{\text{ref}}(\cdot|\mathbf{x})} \left[ \frac{\pi_\theta(\mathbf{y}^w,\mathbf{y}^l|\mathbf{x})}{\pi_{\text{ref}}(\mathbf{y}^w,\mathbf{y}^l|\mathbf{x})} \nabla_\theta \left( \log \frac{\pi_\theta(\mathbf{y}^w|\mathbf{x})}{\pi_{\text{ref}}(\mathbf{y}^w|\mathbf{x}} - \log \frac{\pi_\theta(\mathbf{y}^l|\mathbf{x}}{\pi_{\text{ref}}(\mathbf{y}^l|\mathbf{x}} \right)^2 \right]$$

$$= \mathbb{E}_{\mathbf{x}\sim\mathcal{D},(\mathbf{y}^w,\mathbf{y}^l)\sim\pi_\theta(\cdot|\mathbf{x})} \left[ \left( \log \frac{\pi_\theta(\mathbf{y}^w|\mathbf{x})}{\pi_{\text{ref}}(\mathbf{y}^w|\mathbf{x})} - \log \frac{\pi_\theta(\mathbf{y}^l|\mathbf{x})}{\pi_{\text{ref}}(\mathbf{y}^l|\mathbf{x})} \right) \left( \nabla_\theta \log \pi_\theta(\mathbf{y}^w|\mathbf{x}) - \nabla_\theta \log \pi_\theta(\mathbf{y}^l|\mathbf{x}) \right) \right]$$

$$= \mathbb{E}_{\mathbf{x}\sim\mathcal{D},(\mathbf{y}^w,\mathbf{y}^l)\sim\pi_\theta(\cdot|\mathbf{x})} \left[ \log \frac{\pi_\theta(\mathbf{y}^w|\mathbf{x})}{\pi_{\text{ref}}(\mathbf{y}^w|\mathbf{x})} \nabla_\theta \log \pi_\theta(\mathbf{y}^w|\mathbf{x}) + \log \frac{\pi_\theta(\mathbf{y}^l|\mathbf{x})}{\pi_{\text{ref}}(\mathbf{y}^l|\mathbf{x})} \nabla_\theta \log \pi_\theta(\mathbf{y}^l|\mathbf{x}) \right]$$

$$= \mathbb{E}_{\mathbf{x}\sim\mathcal{D},\mathbf{y}\sim\pi_\theta(\cdot|\mathbf{x})} \left[ \log \frac{\pi_\theta(\mathbf{y}|\mathbf{x})}{\pi_{\text{ref}}(\mathbf{y}|\mathbf{x})} \nabla_\theta \log \pi_\theta(\mathbf{y}|\mathbf{x}) \right] = \mathbb{E}_{\mathbf{x}\sim\mathcal{D}} \left[ \nabla_\theta \text{KL}(\pi_\theta||\pi_{\text{ref}}) \right]$$

Which concludes the proof.

## B.2 Detailed derivation of regularization effect in DAAs

$$\mathbb{E}_{(\mathbf{x},\mathbf{y}^w,\mathbf{y}^l)\sim(\mathcal{D},\pi_{\text{ref}})}\left[\nabla_\theta\frac{1}{2}\rho_\theta^2\right]$$

$$=\mathbb{E}_{(\mathbf{x},\mathbf{y}_1,\mathbf{y}_2)\sim(\mathcal{D},\pi_{\text{ref}})}\left[\nabla_\theta\frac{1}{2}\rho_\theta^2\right]$$

$$=2\mathbb{E}_{(\mathbf{x},\mathbf{y})\sim(D,\pi_{\text{ref}})}\left[\log\frac{\pi_\theta(\mathbf{y}|\mathbf{x})}{\pi_{\text{ref}}(\mathbf{y}|\mathbf{x})}\nabla\log\pi_\theta(\mathbf{y}|\mathbf{x})\right]-$$

$$2\mathbb{E}_{(\mathbf{x},\mathbf{y})\sim(D,\pi_{\text{ref}})}\left[\log\frac{\pi_\theta(\mathbf{y}|\mathbf{x})}{\pi_{\text{ref}}(\mathbf{y}|\mathbf{x})}\right]\mathbb{E}_{(\mathbf{x},\mathbf{y})\sim(D,\pi_{\text{ref}})}\left[\nabla\log\pi_\theta(\mathbf{y}|\mathbf{x})\right]$$

$$=2\mathbb{E}_{(\mathbf{x},\mathbf{y})\sim(D,\pi_{\text{ref}})}\left[\log\frac{\pi_\theta(\mathbf{y}|\mathbf{x})}{\pi_{\text{ref}}(\mathbf{y}|\mathbf{x})}\nabla\log\pi_\theta(\mathbf{y}|\mathbf{x})\right]+$$

$$2D_{KL}\left[\pi_{\text{ref}}(\cdot|\mathbf{x})|\pi_\theta(\cdot|\mathbf{x})\right]\mathbb{E}_{(\mathbf{x},\mathbf{y})\sim(D,\pi_{\text{ref}})}\left[\nabla\log\pi_\theta(\mathbf{y}|\mathbf{x})\right]$$

## C  Training Details

```
For the following dialogue history to a chatbot, which response is more
helpful?

Dialogue history:
<dialogue history>

Response A:
<Response A>

Response B: <Response B>

FIRST provide a one-sentence comparison of the two responses and explain
which you feel is more helpful.  SECOND, on a new line, state only "A" or
"B" to indicate which response is more helpful.  Your response should use
the format:
Comparison:  <one-sentence comparison and explanation>
More helpful:  <"A" or "B">
```

Table 2: Prompt for GPT-4 evaluation on the dialogue generation task. Texts in blue are placeholders to be substituted by the real data.

**SFT Training** For the summarization task, we use the SFT split of Reddit TL;DR summarization. For Anthropic-HH we use the chosen responses from the preference dataset for SFT stage. We pool together both datasets into a single SFT dataset.

**Preference Training** For TL;DR summarization dataset, we train all methods for 2 epochs. To evaluate the efficiency of addressing the over-optimization problem, we vary the regularization strength $\beta\in\{0.01,0.05,0.1\}$.

Across all SFT and Preference training, we use a global batch size of 64 (with 4 gradient accumulation steps), and AdamW optimizer with a learning rate of $1\times10^{-6}$ (cosine learning rate scheduler warm-up for 100 steps) and a max length of 640.

**Golden Reward Training details** We follow the synthetic setup where the *golden* reward model serves as human evaluation and provides preference labels [Gao et al., 2022, Tang et al., 2024a].

We first initialize the golden reward model with a SFT version of Llama-3.1-8B on the pooled SFT data. We then train the golden reward model on the combined preference of the TL;DR and Anthropic-HH dataset. Specifically, we use a batch size of 128, with a learning rate of $5\times10^{-6}$, we train for one epoch with a cosine learning rate schedule.

The golden reward model achieves high validation accuracies with 75.2% validation accuracy, showing high correlation with human preferences.

**Details of KL Divergence Estimation** In this paper, we construct an unbiased estimate of $\text{KL}(\pi_\theta||\pi_{\text{ref}})$ by sampling. More specifically, we first sample $N$ prompts $\{x^i\}$ from the evaluation set, for each prompt $x^i$, we sample a response $y \sim \pi_\theta(\cdot|x^i)$ from the learned policy $\pi_\theta$. Following [Schulman, 2020], The KL divergence is estimated as follows:

$$\frac{1}{N} \sum_{i=1}^{N} \log \frac{\pi_\theta(y^i|x^i)}{\pi_{\text{ref}}(y^i|x^i)} + \left( \frac{\pi_\theta(y^i|x^i)}{\pi_{\text{ref}}(y^i|x^i)} - 1 \right)$$

**Compute Resources Specification** We train and evaluate our models using NVIDIA 4xH100 GPUs. All evaluations are computed with "gpt-4o-mini" as judge, with random positional flips to avoid position bias.

# D  Evaluation Details

**Golden reward Evaluation** we sample 2 completions per prompt from the learned policy with 512 prompts from the evaluation set. We sample with temperature $\tau = 0.7$ and top-$p$ sampling with $p = 0.95$. to evaluate its performance. To calculate the winrate, we consider all combinations of pairs between the completions generated by the learned policy and the reference completions from the preference dataset, and then compare the scores from the golden reward model on the pair of generations to calculate win-rate.

**GPT-4 Evaluation** For GPT-4 evaluation, we sample 256 prompts from the evaluation set. For each prompt, we sample 1 completion from the learned policy. To evaluate the dialogue generation task, we use the prompt shown in Table 2, similar to [Ji et al., 2024] with random position flipping to avoid position bias.

# E  IS-DPO as an approximation of Online-DPO

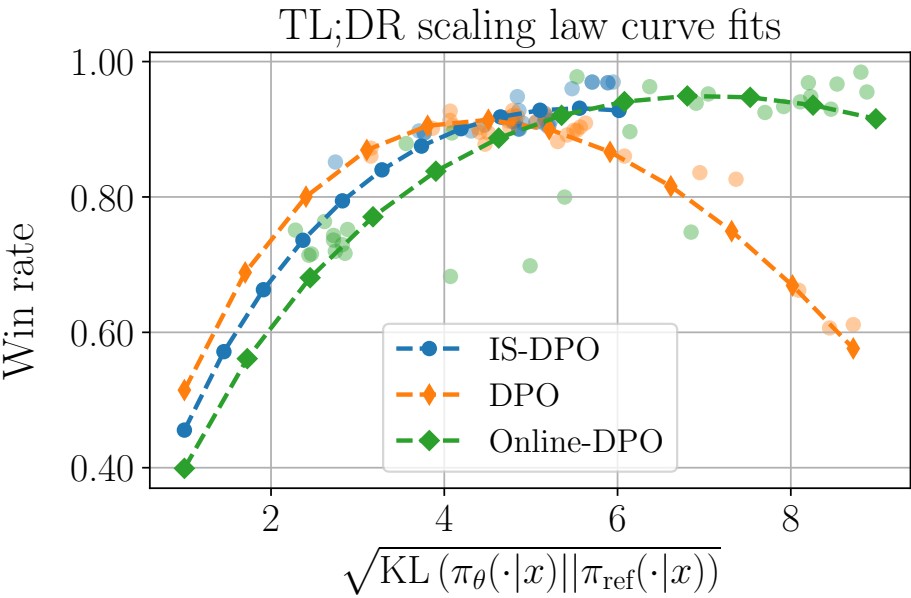

Figure 7: Scaling law curve fits between DPO, IS-DPO and Online-DPO

We have conducted experiments to understand how well the IS-DPO can better approximate the Online-DPO Guo et al. [2024]. Our findings are below:

**IS-DPO can closely approximate Online-DPO up to a specific KL divergence threshold:** We observe that IS-DPO also follows a similar performance trajectory to Online-DPO in the low to moderate KL divergence regime . In this region, both methods show a similar and steady increase in win rate. This suggests that IS-DPO is capable of closely approximating the behavior and performance of Online-DPO within a certain KL "budget".

**On-policy benefits from large KL "budget":** While IS-DPO does not show signs of over-optimization, it also does not extrapolate to higher KL regimes, while Online-DPO can find better policies under higher KL regimes. This indicates that on-policy optimization remains beneficial when the optimal policies lie farther from the , as Online-DPO is able to leverage a larger KL budget.

**Comparison with DPO:** We also provide a comparison between Online-DPO and DPO. Interestingly, we observe that under a low KL budget, DPO appears to outperform Online-DPO. However, it sharply declines as KL increases, indicating over-optimization, while IS-DPO and DPO effectively utilize a larger KL budget to achieve higher performance.

## F    The role of Importance Sampling in IS-DPO and PPO.

While Importance Sampling (IS) has also been used to address distribution shift in PPO, its usage is fundamentally different from our paper.

PPO uses clipped importance sampling (IS-clip) to improve sample efficiency. Since each batch is typically discarded after a single gradient update in standard policy gradient methods, IS allows PPO to reuse the same batch multiple times, extracting the most information from each batch. In contrast, our objective is to enforce an effective regularization, to prevent them from pushing up the probability of responses that are out of the offline data distribution, mitigating the over-optimization issue. In addition, in PPO, the clipping mechanism acts as a surrogate trust region to prevent the policy from deviating too far from the previous one (which can help mitigate over-optimization), but in a fully offline setting (as in our paper), this mechanism becomes overly conservative-where gradient is zero when the probability ratio falls outside the clipping range, limiting the potential improvement from [Chen et al., 2023, Meng et al., 2023]. Our clipping strategy, conversely, mitigates excessively large updates with IS while still providing meaningful updates even when the ratio lies outside the clipping region.

## G    Theoretical Insight of clipping ratio $\epsilon$ and KL regularization.

**Proposition 2** *Assuming* $supp(\pi_\theta) = supp(\pi_{ref})$, *then the variance of the importance weight* $Var_{\pi_{ref}} \left( \frac{\pi_\theta}{\pi_{ref}} \right)$ *is an upper bound of KL divergence:*

$$KL(\pi_\theta || \pi_{ref}) \leq Var \left( \frac{\pi_\theta}{\pi_{ref}} \right) \tag{16}$$

The proof is straightforward by utilizing the inequality $\log x \leq x - 1, \forall x > 0$. This proposition highlights that when regularization is weak (i.e., small $\beta$), the policy can deviate significantly from $\pi_{ref}$, leading to high variance in the importance weights. This motivates the use of a smaller clipping threshold to control variance and stabilize training under such regimes.

Additionally, we also conduct the relationship between the regularization parameter $\beta$ and the clipping ratio $\epsilon$. We present the win-rate and KL values for TL;DR summarization dataset below: We observe that increasing the clipping threshold allows for larger policy updates, which in turn leads to higher KL divergence across all values of $\beta$. Interestingly, at low regularization levels (small $\beta$), using a smaller clipping ratio yields better performance, as it helps prevent over-optimization. Overall, our experimental results is consistent with Proposition 2, where smaller clipping threshold should be used for small regularization strength to avoid over-optimizaion issue.

## H    Computational cost of calculating importance ratio.

The computational overhead of calculating the importance ratios is negligible and results in almost the same time complexity as DPO. Despite adding the additional importance ratio calculation step to

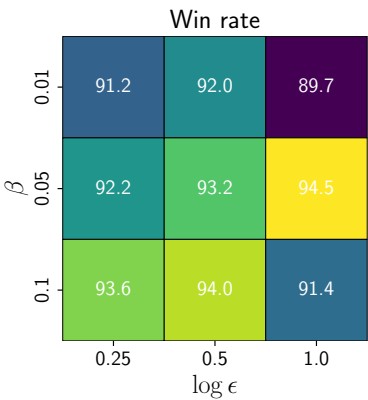
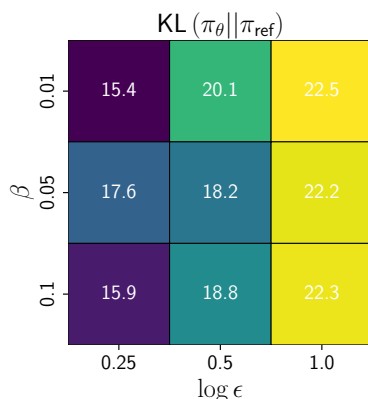

Figure 8: Win rate and KL divergence across different $\epsilon$ and $\beta$.

the original DPO loss, this ratio can be efficiently computed using intermediate quantities already available in the DPO loss calculation. Specifically, as presented in the paper, the importance ratio can be computed in the log-space as follows:

$$w(\mathbf{y}_w, \mathbf{y}_l) = \exp\left(\log \pi_\theta(\mathbf{y}_w) + \log \pi_\theta(\mathbf{y}_l) - \log \pi_{\text{ref}}(\mathbf{y}_l) - \log \pi_{\text{ref}}(\mathbf{y}_w)\right) \tag{17}$$

Hence, IS-DPO only adds a small calculation, negligible compared to the much more heavier computations in the vanilla DPO. This implementation is also straightforward and simple, requiring only a single additional line of code.

To empirically validate this claim, we report the average training time over 100 steps for both DPO and IS-DPO. We use 4 A100-80GB GPUs, a batch size of 16, and LLaMA-3.2-3B as the base model:

| IS-DPO | DPO |
|---|---|
| $92.829\text{s} \pm 5.144$ | $92.571\text{s} \pm 4.837$ |

As can be observed, the training time for IS-DPO is nearly identical to that of the vanilla DPO, confirming that our approach is computationally efficient and negligible in overhead.

## I    The necessity of support constraint in IS-DPO.

While the optimal policy under the DPO objective satisfies:

$$\pi^*(\mathbf{y}|\mathbf{x}) = \frac{1}{Z(\mathbf{x})}\pi_{\text{ref}}(\mathbf{y}|\mathbf{x})\exp(r(\mathbf{x},\mathbf{y})) \tag{18}$$

Thus shares the same support as $\pi_{\text{ref}}$. However, prior works [Rafailov et al., 2024, Azar et al., 2024, Song et al., 2024, Xu et al., 2024] have shown that DAAs assume that $\pi_{\text{ref}}$ has full support over the entire prompt-response space to achieve $\pi^*$. In practice, this assumption rarely holds, as preference datasets only cover a small fraction of the prompt-response space. Consequently, multiple distinct policies can achieve the same global optimum of the DPO objective [Xu et al., 2024, Azar et al., 2024, Rafailov et al., 2024], including policies that assign probability mass to responses outside the support of $\pi_{\text{ref}}$. This contrasts to the original RLHF framework, which prevents generating responses that are of support via explicit KL regularization. Thus, our proposed importance sampling method further enforces the support constraint, helping to avoid this failure mode.

## J    Connection to principle of pessimism in Offline RL.

At a high level, the principle of pessimism explicitly subtracts uncertainty-based penalties such as KL divergence or divergence from the estimated value with the aim of preventing overestimation of the value of candidate policies in regions with low data coverage or high uncertainty.

IS-DPO also incorporates a form of pessimism by assigning lower weights to trajectories that are less likely under the learned policy, effectively penalizing policies that place high probability mass in areas poorly covered by the data. This mechanism aligns conceptually with pessimism-based regularization [Liu et al., 2024a, Zhu et al., 2024]. Moreover, there is a deeper theoretical connection worth exploring: the variance of the IS estimator $\text{Var}_{\pi_{\text{ref}}}\left(\frac{\pi_\theta}{\pi_{\text{ref}}}\right)$ is proportional to the chi-squared divergence between $\chi^2(\pi_\theta, \pi_{\text{ref}})$ This observation suggests a promising direction to formalize the connection between importance sampling and pessimism. $\chi^2$ divergence has been used in LLM alignment to impose stronger regularization, thereby effectively mitigating over-optimization issues. We believe that bridging this connection more formally, possibly through uncertainty-aware confidence intervals, is a promising avenue for future work.

