# OpenReview forum: "Mitigating Reward Over-optimization in Direct Alignment Algorithms with Importance Sampling"
_NeurIPS.cc/2025/Conference — NeurIPS 2025 poster_

### Official Review · Reviewer_VYu7 · 2025-07-02

**Clarity:** 3
**Significance:** 3
**Originality:** 3
**Rating:** 5
**Confidence:** 4

**Summary:**

This paper proposes added importance weights to the DPO objective to mitigate over-optimization. This is simply done by multiplying the loss of each example by the density ratio of each sample under the current vs reference policy.

**Questions:**

1. Why does IS-DPO seem to be invariant to $\beta$? It appears from Figure 4 that no matter the $\beta$, IS-DPO seems to converge to a sqrt(KL) of 5, while other methods seem to change. Was there possibly a bug here?
2. Why do the authors choose 2 epochs for training? Is one epoch not more standard for DPO? Would this not increase the overfitting effect in Fig 3?
3. What is the effect of SFT on IS-DPO, given that without it, the importance weights are incorrect?
4. What values of $\beta$ were used for figure 3?

**Ethical Concerns:**

["NO or VERY MINOR ethics concerns only"]

**Final Justification:**

My question about the KL-budget experiment seems answered. The authors also clarified a number of points from the draft. I left a comment listing some improvements that I think should be made, namely that Sec. 4.3 should be explained a bit more (tying into the KL budget experiments) and that the SFT assumption should be further clarified.

Generally, I recommend accepting the paper as its results seem relatively good given a small overhead, and is justified adequately.

**Limitations:**

No limitations section is provided -- it would be good to include one!

**Quality:**

3

**Strengths And Weaknesses:**

## Strengths
* The method is simple to implement
* The approach appears to improve performance with minimal changes.
* The approach is well motivated and explained. I particularly liked the examples in Fig 1 and Sec. 3.2.
* The paper is sufficiently self-contained -- one can get the required background from reading the work.
* the chosen baselines appear sufficient to me and the paper focuses on direct alignment algorithms. The paper could be made stronger by considering other baselines outside of DAAs (e.g. GRPO), but I concede that it seems a bit orthogonal at present.


## Weaknesses

**Theory and Presentation**


* Some of the theory presented is a bit confusing. Why should we consider the taylor expansion of the gradient as in Tang et al. when we could just directly consider the gradient? In the gradient itself (I believe) we do not get this KL term from eq. 8. Moreover, the KL term is only valid at the start of training, and progressively goes away with more updates. To summarize, I am not certain if I understand why the argument from Tang et al. logically holds here.

* Missing assumptions: DAAs do not necessarily require that the policy which produced D is the same as the reference policy $\pi_\text{ref}$ -- they can operate over any support. However it has been implicitly assumed by the authors via the importance weights that the sampled responses do come from the reference policy. It seems necessary to highlight this, and point out that it means that SFT on the preference dataset is likely required for IS-DPO to work.

* Theorem 1 is kinda just saying that importance sampling is unbiased.... which is not necessarily a new result and I am unsure if it deserves to be a theorem. It is also strange to state "the gradient concerning $\pi_\theta$ of the square loss is equal to that of the KL when this is again another straightforward application of importance sampling and only holds when $\rho_\theta = 0$ at the start of training. That second component should be mentioned again.

* In section 4.3 it is asserted that when supp(P) is a subset of supp(Q), the Q term completely disappears? How do the authors go from P(x)/Q(x) --> P(x)? Would this not instead just be undefined? I thought importance sampling would require the support constraint. This argument was also a bit confusing because a fixed dataset (assumed to be sampled from piref) is used, and if $\pi^* \propto  \pi_\text{ref} e^r$, then the support of $pi^*$ will be the same as that of pi ref unless the reward function is unbounded.

**Experiments**
* the experimental results are a bit limited in comparison to several other modern LLM papers, however I understand computational constraints are a factor. HH and TLDR are rather simple datasets at this point.
* the gains, though positive, are a bit marginal in many cases.
* The effect on KL evolution appears to be inconsistent (see question).
* While IS-DPO perhaps exhibits less over-optimization in Fig 3, its performance at different KL budgets appears to be worse?

Minor:
* missing reference on line 39
* should note that $\sigma$ is being used as the logistic function.
* Line 123 "going in the opposite direction of the second term" is ambiguous
* Line 254 missing reference
* Line 243 missing reference
* Line 223: previous approaches *are*
The authors should spend some time reading the paper to catch these issues.

## Overall
Overall, I recommend a borderline accept. The paper has a reasonable idea -- that we can use importance weights to bring offline DAAs closer to online methods. However, it is hard to fully recommend the paper because:
1. The assumptions of the method and theory are not explicitly stated, and parts of the text around this are confusing.
2. The experimental results are a little inconsistent (KL evolutions and performance in Fig 3 at fixed budgets)

Should the authors be able to better explain the theory results and inconsistencies in experiments I am willing to raise my score.

---

> ### Author Rebuttal · Authors · 2025-07-31
>
> Thank you for your review and positive feedback on our paper. We appreciate your acknowledgment of the clarity and organization of our work. Please find our response to your comments below.
>
> **Q1. Why should we consider the Taylor expansion? And the KL term is only valid at the start of training**
>
> **Answer:** Thanks for the great question. You are correct that directly computing the gradient does not yield the KL term as shown in Equation 8. In Section 2, our goal is to highlight that the KL regularization term is only meaningful at the beginning of training when using DAAs. Without importance sampling or on-policy sampling, this KL term in DAAs fails to accurately reflect the KL regularization used in RLHF as training progresses with more updates. Hence, by multiplying with the importance ratio, we can enforce more effective regularization and better reflect the KL regularization in RLHF without requiring on-policy samples.
>
> **Q2. Missing assumptions: DAAs do not necessarily require $\pi_{\text{ref}}$ to produce $\mathcal D$ and the necessity of SFT for IS-DPO.**
>
> **Answer**: Thank you for pointing this out. We first apologize for the implicit assumption in our formulation that the policy produced $\mathcal D$ is the same as $\pi_{\text{ref}}$. We will make it more explicit in the final version.
>
> Thank you for a great comment on the necessity of (SFT) on the preference dataset for IS-DPO to be effective. We'd like to clarify that this requirement is not unique to IS-DPO but is also essential for DPO and other DAAs [11,12].
>
> Nevertheless, we agree that the suggestion of the reviewer can be interesting, thus we also provide experiments exploring how both DPO and IS-DPO perform when trained on preference data not aligned with $\pi_{\text{ref}}$. Specifically, we construct a new dataset $\mathcal{D}\_{\text{shift}}$ by sampling responses from final Online-DPO policies, which achieve a high win rate of $\approx 97\%$ but exhibit a significant shift from $\pi_{\text{ref}}$ $\left(\sqrt{\text{KL}(\pi_\theta || \pi_{\text{ref}})} \approx 7.2\right)$. These responses are then labeled using the golden reward to generate 100k preference samples. We train both IS-DPO and DPO on $\mathcal{D}_{\text{shift}}$ with $\beta=0.05$ and $\log\epsilon=0.5$. We present our results below:
>
> | Method | Win-rate | $\sqrt{\text{KL}}$ |
> | -------- | -------- | -------- |
> | DPO     | 74.6     | 7.2     |
> | IS-DPO | 77.3 | 3.86
>
> Both methods achieve lower win rates compared to training on the original dataset, reaffirming the importance of SFT as a crucial warm-up step. Interestingly, IS-DPO achieves a slightly higher win rate than DPO while maintaining significantly lower KL divergence from the $\pi_{\text{ref}}$. This suggests that IS-DPO may exhibit more robust to distribution shifts even when preference data is not sampled from $\pi_{\text{ref}}$.
>
> **Q3. Theorem 1 just says that importance sampling is an unbiased estimator, which is not a new result, and strange statement.**
>
> **Answer:** Thank you for your suggestions. While the unbiased nature of IS is not new, this theorem, and our paper in general, formalize the impact of distribution shift in DAAs and provide a theoretical analysis (see Section 3.1) to justify the IS correction under DPO objective. Specifically, we show that minimizing the implicit regularization in DPO when applied with importance sampling can also minimize the KL divergence. This connection not only grounds our method in addressing distribution shift but also provides a crucial link between our proposed regularization and KL regularization in RLHF via Theorem 1, as acknowledged by reviewer PaV2. Nevertheless, we will consider the reviewer's suggestion and revise the theorem accordingly to reflect what's known and what's our focus (as in this discussion).
>
> **Q4. The support constraint in Importance Sampling (Sec 4.3)**
>
> **Answer:** Thank you for your question. To further elaborate, we consider the support of a discrete distribution $P$ over sample space $\mathcal X$ as follows: $\text{supp}(P)=\{x\in\mathcal X|P(x) > 0\}$. The IS estimator is defined as $\mathbb E_{x\sim Q}\left[\frac{P(x)}{Q(x)}f(x)\right]=\sum\limits_{x\in\text{supp}(Q)}Q(x)\frac{P(x)}{Q(x)}f(x)=\sum\limits_{x\in\text{supp}(Q)}P(x) f(x)$. When we assume $\text{supp}(P) \subseteq \text{supp}(Q)$, we can see that the IS estimator is unbiased, $\sum\limits_{x\in\text{supp}(Q)}P(x) f(x) =\sum_{x\in\text{supp}(P)}P(x) f(x)=\mathbb{E}_{x\sim P}\left[f(x)\right]$.
>
> While it's true that the optimal policy under the DPO objective satisfies $\pi^* \propto \pi_{\text{ref}} e^{r}$, and thus shares the same support as $\pi_{\text{ref}}$. However, prior works [1,2,3,4] have shown that DAAs assume that $\pi_{\text{ref}}$ has full support over the entire prompt-response space to achieve $\pi^*$. In practice, this assumption rarely holds, as preference datasets only cover a small fraction of the prompt-response space. Consequently, multiple distinct policies can achieve the same global optimum of the DPO objective [2,3,4], including policies that assign probability mass to responses outside the support of $\pi_{\text{ref}}$ [1, 3]. This contrasts to the original RLHF framework, which prevents generating responses that are of support $\pi_{\text{ref}}$ via explicit KL regularization. Thus, our proposed importance sampling method further enforces the support constraint, helping to avoid this failure mode.
>
> **Q5. Limited experiments with HH and TLDR are simple datasets.**
>
> **Answer:** Thank you for your feedback. We employ HH and TLDR, as they are widely adopted standard benchmarks in the alignment literature [4,5,6,7] for evaluation of DPO and its variants. Consequently, our work should be evaluated on these benchmarks for a fair evaluation with the existing baselines; evaluating on additional datasets should indeed incur significant computational constraints.
>
> Nevertheless, please note that we have also conducted additional experiments on the UltraFeedback dataset in the paper, where we evaluate IS-DPO using popular benchmarks such as AlpacaEval 2.0 and MT-Bench, designed to measure alignment with human preferences in more practical settings. These results (Section 4.2) further support the effectiveness of IS-DPO beyond HH and TLDR. We hope this extended evaluation addresses your concerns regarding the scope of our experimental validation.
>
> **Q6. The invariance of IS-DPO to $\beta$.**
>
> **Answer:** Thank you for your great comment. We confirm that there is no bug in Fig. 4 of IS-DPO. We observed that $\text{KL}(\pi_\theta,|, \pi_{\text{ref}})$ during training typically remains within the range of 22.87 to 27.06. As discussed in Section 4.3, this behavior arises because the importance ratio in IS-DPO gradually decreases throughout training. As the importance weights diminish, resulting in an implicit early stopping effect.
>
> This mechanism provides two key benefits. First, a low importance ratio implies that the current policy $\pi_\theta$ assigns very low likelihood to certain responses, and further updating on such samples can lead to over-optimization [5, 9]. IS-DPO naturally avoids this issue by down-weighting such updates. Second, a low importance weight may signal that on-policy sampling is needed, which is especially beneficial in iterative learning settings [10].
>
> **Q7.  Why 2 epochs for training instead of 1 epoch?**
>
> **Answer:** Thank you for your question. Prior work [9] has shown that DPO can suffer from over-optimization even before completing a single epoch, with further training resulting in performance degradation. Motivated by this, our work focuses on settings where over-optimization is a particular concern. In the Controlled Summarization task, we deliberately train for 2 epochs to highlight the robustness of IS-DPO under prolonged training and weak regularization (i.e., low $\beta$); this setting is also consistent with prior studies [5, 6]. To ensure fair comparisons with standard setups, we still follow the standard 1-epoch configuration used in Instruction Following tasks [4, 7, 8] to demonstrate IS-DPO effectiveness in aligning with human preferences.
>
> **Q8. The values of $\beta$ in Fig. 3? Why IS-DPO exhibit worse performance at different KL budgets**
> **Answer:** Thank you for your questions. We consider a range of $\beta\in\{0.01, 0.05, 0.1\}$ to consider different KL values in Fig 3. Viewing KL as a resource to be spent, we observe that while DPO exhibits better performance under a low KL budget, IS-DPO tends to consume more of the effective KL budget, ultimately achieving a higher peak win rate. Interestingly, we found that this phenomenon also exhibits in Online-DPO and previous work [8,13].
>
> ----
>     [1] Is DPO Superior to PPO for LLM Alignment? A Comprehensive Study. ICML24 Oral.
>     [2] A General Theoretical Paradigm to Understand Learning from Human Preferences. AISTATS24.
>     [3] The Importance of Online Data: Understanding Preference Fine-tuning via Coverage. NeurIPS24.
>     [4] Scaling Laws for Reward Model Overoptimization in Direct Alignment Algorithms. NeurIPS24.
>     [5] Disentangling Length from Quality in Direct Preference Optimization. ACL24.
>     [6] Correcting the Mythos of KL-Regularization: Direct Alignment without Overoptimization via Chi-Squared Preference Optimization. ICLR24 Spotlight.
>     [7] REBEL: Reinforcement learning via regressing relative rewards. NeurIPS24.
>     [8] Direct Language Model Alignment from Online AI Feedback. CORR24
>     [9] Learning Dynamics of LLM Finetuning. ICLR25 Oral.
>     [10] Direct Nash Optimization: Teaching Language Models to Self-Improve with General Preferences. CoRR24.
>     [11] Direct Preference Optimization: Your Language Model is Secretly a Reward Model. NeurIPS23 Oral.
>     [12] ORPO: Monolithic Preference Optimization without Reference Model. ACL24.
>     [13] Understanding the performance gap between online and offline alignment algorithms. CoRR24

---

> > ### Comment · Reviewer_VYu7 · 2025-08-05
> > **Thanks for the response**
> >
> > Thanks for the response, some of my concerns have been addressed so I'll raise my score accordingly. I think there are still a few things the authors should incorporate into their final version:
> >
> > * One point I would push back on is the SFT requirement. While it is true that DAAs generally perform better with it, there is nothing *mathematical* that necessitates SFT for DPO. However, this is true for for IS-DPO, which I think is an important distinction that should be made.
> > * Having an appendix section that expands 4.3 a bit seems like it would be helpful -- the authors response was a good clarification here, and also a good place to discuss the KL budget behavior that looks very different than other methods.

---

> > > ### Author Response · Authors · 2025-08-06
> > > **Thanks for your reply.**
> > >
> > > Thank you very much for your reply. We will further improve the paper in the final camera-ready version based on your suggestions!

---

### Official Review · Reviewer_PaV2 · 2025-07-03

**Clarity:** 3
**Significance:** 3
**Originality:** 2
**Rating:** 4
**Confidence:** 4

**Summary:**

This paper highlights the problem of reward over-optimization in Direct Alignment Algorithms (DAAs) such as DPO and IPO, which occurs when models diverge excessively from the reference policy during alignment, leading to suboptimal performance. To mitigate this, the authors propose an adaptive low-variance importance sampling (IS) strategy for offline post-training alignment of large language models (LLMs). Empirical evaluations demonstrate the benefit of the above strategy via mitigating reward over-optimization.

**Questions:**

Please see weakness section

**Ethical Concerns:**

["NO or VERY MINOR ethics concerns only"]

**Final Justification:**

I keep my scores towards acceptance. There were revisions, but after incorporating them, i believe it can be accepted

**Limitations:**

Please see weakness section

**Quality:**

3

**Strengths And Weaknesses:**

The paper uses importance sampling (IS) and intelligent modifications to approximate expectations under the learned policy and mitigate distribution shift. While IS is well-known in the literature, its application here is intelligent and appropriate. It also shows that the regularization can be connected to the KL penalty in RLHF through Theorem 1, which is crucial—though it may require further justification. The work establishes a strong link between loss curvature and KL regularization in RLHF. Empirical analysis shows it outperforms baselines such as RPO, length regularization, and others.

Although the paper proposes an interesting approach, certain claims need further clarity. A key issue is regarding the KL regularization to the reference policy, which is a crucial component in the alignment objective. It is not extremely clear in the scenario where there is clipped importance weight being used, how the KL is still ensured mathematically w.r.t the original alignment objective? Theorem 1 provides the details, but an explicit description showing how the importance weight affects the KL balls is crucial? Empirical analysis is also crucial
Second, does the unbiasedness of the gradient estimate still hold true with the importance weight and smoothening? It will be helpful to highlight the same. In the importance weight estimation, there can be trajectories generated whose probability is low under the reference policy making the weights unbounded, how the smoothing helps in this regard? Additionally, the Taylor approximation is local and mainly valid around 0, i.e when the policies are close to reference. However, how this helps in scenarios when they are apart which is the main reason of the dist shift issue leading to overoptimization, then the approximation might not hold. Can you provide a detailed discussion regarding the same? Discussion with works that rely on the principles of pessimism needs to be discussed [1,2].

References
[1]. Iterative data smoothing: Mitigating reward overfitting and overoptimization in rlhf
[2]. Provably Mitigating Overoptimization in RLHF: Your SFT Loss is Implicitly an Adversarial Regularizer

---

> ### Author Rebuttal · Authors · 2025-07-31
>
> Thank you for your great comments and efforts in reviewing our paper. Please find our response to your comments below.
>
> **Q1. A key issue is regarding the KL regularization to the reference policy, which is a crucial component in the alignment objective. It is not extremely clear in the scenario where there is clipped importance weight being used, how the KL is still ensured mathematically w.r.t the original alignment objective? Theorem 1 provides the details, but an explicit description showing how the importance weight affects the KL balls is crucial? Empirical analysis is also crucial**
>
> **Answer:** Thank you for your question. We acknowledge that Theorem 1 only provides the details for importance weight. Below, we provide an explanation to clarify how the clipped importance weight affects alignment with the KL regularization. We consider the $\mu$-squared loss with importance sampling:
> $$\nabla_\theta\mu_{\text{IS}}(y_1,y_2)=w\nabla_\theta\rho, \quad \nabla_\theta\mu_{\text{clip}}(y_1,y_2)=\min(w, c)\nabla_\theta \rho$$
> where $w=\frac{\pi_\theta(y_1,y_2)}{\pi_{\text{ref}}(y_1,y_2)},\rho=\log\frac{\pi_\theta(y_1)}{\pi_{\text{ref}(y_1)}}-\log\frac{\pi_\theta(y_2}{\pi_{\text{ref}(y_2)}}$. As shows in Theorem 1, we can see that $\mathbb E_{\pi_{\text{ref}}}[\nabla_\theta\mu_{\text{IS}}]=\nabla_\theta\text{KL}(\pi_\theta||\pi_{\text{ref}})$.
>
> **Clipped IS-DPO ensures alignment with KL regularization**: Notice that the gradient of the clipped objective can be expressed as
>
> $$\\nabla\_{\\theta}\mu\_{\text{clip}}=\nabla\_\theta\mu\_{\text{IS}}-\mathrm{1}\_{\{w > c\}}(w-c)\nabla\_\theta\rho$$
>
> This decomposition shows that clipping introduces a bias by down-weighting the influence of samples with large importance weights. However, the clipped gradient still maintains alignment with the importance-weighted gradient, as indicated by the dot product.
> $$\langle\nabla_\theta\mu_{\text{clip}}, \nabla_\theta\mu_{\text{IS}}\rangle \geq 0$$
>
> In effect, clipping acts as a form of regularization that mitigates gradient explosion from high-weight outliers while preserving alignment with the KL regularization.
>
> **Q2. Does the unbiasedness of the gradient estimate still hold true with the importance weight and smoothening? It will be helpful to highlight the same. In the importance weight estimation, there can be trajectories generated whose probability is low under the reference policy making the weights unbounded, how the smoothing helps in this regard?**
>
> **Answer:** We really appreciate this question. We sincerely apologize for the typo in Equation (13) — the operator should indeed be $\min$ instead of $\max$.
>
> While the clipping operator introduces into the estimator, its purpose is to keep the gradient variance under control. This clipping mechanism is especially important in cases where certain trajectories have very low probability under $\pi_{\text{ref}}$. The clipping mechanism will be applied the keep the importance weight bounded to stabilize training.
>
> **Q3. Additionally, the Taylor approximation is local and mainly valid around 0, i.e, when the policies are close to reference. However, how this helps in scenarios when they are apart which is the main reason of the dist shift issue leading to overoptimization, then the approximation might not hold. Can you provide a detailed discussion regarding the same?**
>
> **Answer:** Thank you for your question. You are correct that the Taylor approximation used in Offline DAAs only enforces KL regularization when the policies $\pi_\theta$ are close to $\pi_{\text{ref}}$. This is the exact objective our our analysis in this section: we aim to show that existing DAAs will only have effective regularization locally (via this Taylor expansion) and become susceptible to overoptimization as the learned policy diverges further.
>
> More specifically, this approximation shows that DAAs are not well regularized when the learned policy $\pi_\theta$ moves away from $\pi_{\text{ref}}$. Optimizing Offline DAAs can lead to over-optimization or extrapolation into out-of-support regions of $\pi_{\text{ref}}$ $\left(\text{KL}(\pi_\theta||\pi_{\text{ref}}=\infty\right)$ [3,4,5,6], and the local regularization used in Offline DAAs is insufficient to prevent these issues compared to standard RLHF.
>
> **Q4. Discussion with works that rely on the principles of pessimism needs to be discussed [1,2].**
>
> **Answer** Thank you for the insightful suggestion. We will include the following discussion, citing the works on principles of pessimism, in the camera-ready version:
>
> At a high level, the *principle of pessimism* explicitly subtracts uncertainty-based penalties such as KL divergence or $\chi^2$ divergence from the estimated value with the aim of preventing overestimation of the value of candidate policies in regions with low data coverage or high uncertainty.
>
> IS-DPO(IS) also incorporates a form of pessimism. IS-DPO assigns lower weights to trajectories that are less likely under the learned policy $\pi_\theta$, effectively penalizing policies that place high probability mass in areas poorly covered by the data. This mechanism aligns conceptually with pessimism-based regularization [1,2].
>
> Moreover, there is a deeper theoretical connection worth exploring: the variance of the IS estimator is proportional to the chi-squared divergence between $\pi_\theta$ and $\pi_{\text{ref}}$. This observation suggests a promising direction to formalize the connection between importance sampling and pessimism. $\chi^2$ divergence has been used in LLM alignment to impose stronger regularization, thereby effectively mitigating over-optimization issues. We believe that bridging this connection more formally, possibly through uncertainty-aware confidence intervals, is a promising avenue for future work.
>
> ----
>     [1] Iterative Data Smoothing: Mitigating Reward Overfitting and Overoptimization in RLHF. ICML24.
>     [2] Provably Mitigating Overoptimization in RLHF: Your SFT Loss is Implicitly an Adversarial Regularizer. NeurIPS24.
>     [3] Is DPO Superior to PPO for LLM Alignment? A Comprehensive Study. ICML24 Oral.
>     [4] A General Theoretical Paradigm to Understand Learning from Human Preferences. AISTATS24.
>     [5] The Importance of Online Data: Understanding Preference Fine-tuning via Coverage. NeurIPS24.
>     [6] Scaling Laws for Reward Model Overoptimization in Direct Alignment Algorithms. NeurIPS24.
>     [7] Correcting the Mythos of KL-Regularization: Direct Alignment without Overoptimization via Chi-Squared Preference Optimization. ICLR25 Spotlight.

---

> > ### Comment · Reviewer_PaV2 · 2025-08-02
> > **Response to Rebuttal by Authors**
> >
> > Thanks for the detailed clarification and I am happy to keep the score, leaning towards acceptance.

---

### Official Review · Reviewer_B49u · 2025-07-03

**Clarity:** 3
**Significance:** 3
**Originality:** 2
**Rating:** 4
**Confidence:** 2

**Summary:**

This paper addresses the problem of reward over-optimization in offline Direct Alignment Algorithms (DAAs) like DPO. The authors identify that DAA's implicit KL regularization is only effective when the learned policy is close to the initial reference policy, leading to performance degradation as the model drifts away during training. To mitigate this, the paper proposes Importance Sampling DAAs IS-DAAs, a novel approach that re-weights the standard DAA loss with an importance sampling ratio. This re-weighting aims to better approximate the online learning objective using only offline data, thus enforcing a more effective regularization throughout the training process. To handle the high variance typically associated with importance sampling, the authors propose clipping the importance weights. The paper provides theoretical analysis showing that their method provides an unbiased estimate of the online DPO objective and that its gradient correctly corresponds to the gradient of the KL divergence. Through experiments on TL;DR summarization and instruction-following benchmarks, the authors demonstrate that IS-DAA effectively mitigates over-optimization, particularly under low regularization, and achieves superior performance compared to standard DAA and other baseline methods designed to tackle the same problem.

**Questions:**

1. Can the authors offer a hypothesis or analysis for why X-PO might be more effective on AlpacaEval, while IS-DPO is stronger on Anthropic-HH? X-PO uses a chi-squared divergence which penalizes large deviations from the reference policy more heavily than the KL divergence implicit in DPO. Does IS-DPO, by more faithfully approximating an on-policy KL-regularized objective, perhaps inherit a specific behavior that is less suited to the data distribution or evaluation protocol of AlpacaEval?
2. Could the authors provide more practical guidance on setting $\epsilon$? A key question is its interaction with the main DAA regularization parameter $\beta$. How should one co-tune $\beta$ and $\epsilon$? For instance, if one uses a very high $\beta$ (forcing the policy to stay close to the reference), does the optimal $\epsilon$ tend to be larger or smaller? Does the optimal $\epsilon$ depend on properties of the offline dataset, such as its size or the diversity of the reference policy responses?

**Ethical Concerns:**

["NO or VERY MINOR ethics concerns only"]

**Final Justification:**

My major concern on the novelty of the proposed clipping importance-sampling technique is well addressed. Therefore, I raised my score accordingly.

**Limitations:**

Yes.

**Paper Formatting Concerns:**

None.

**Quality:**

2

**Strengths And Weaknesses:**

Strengths:

1. Significance: The paper tackles a well-known and highly relevant problem. Reward over-optimization and the performance gap between offline DAAs and online RLHF are significant barriers to creating more robust and reliable aligned models. The community is actively seeking simple and effective solutions, and this paper proposes a strong candidate.
2. Clarity: The paper is well-written and easy to follow. The authors clearly motivate the problem, starting from the limitations of RLHF, moving to the promise and pitfalls of DAAs, and culminating in a crisp diagnosis of why their implicit regularization fails (Section 2.3). The progression from the standard DAA objective to the proposed IS-DAA objective is logical and well-explained.

Weaknesses:

1. Originality: The primary weakness is the novelty of the proposed technical components. As noted, importance sampling with clipping is proposed in PPO and serves as a cornerstone of modern RL. The paper re-applies this established technique in the context of DAAs. An expert in RL would immediately recognize the objective in Eq. (11) and the clipping in Eq. (13) as analogous to PPO's surrogate objective. The authors could be more explicit about this connection to prior work such as PPO. The contribution is therefore not in inventing a new technique, but in the application, diagnosis, and analysis of an existing one in a new domain.
2. Quality: The paper introduces a clipping hyperparameter to control the bias-variance trade-off. However, the analysis is limited to a single ablation study (Figure 5). There is little discussion on the sensitivity to this hyperparameter or practical guidance on how to set it. Since clipping fundamentally changes the objective from an unbiased to a biased one, the paper could benefits from a deeper discussion of its impact and how it might interact with the learning rate or the DAA's own parameter.
3. Quality: The method is motivated by better approximating the online DAA objective. A powerful, albeit computationally expensive, experiment would have been to compare IS-DAA (the offline approximation) directly against an online DPO baseline (which performs actual on-policy sampling). This would have provided a clear measure of how successfully the importance sampling bridges the gap between offline and online training.

---

> ### Author Rebuttal · Authors · 2025-07-31
>
> We thank the reviewer for their helpful feedback and will respond to the raised questions below.
>
> **Q1. Discussion between the connection of clipped Importance sampling in IS-DPO and PPO.**
>
> **Answer:** Thank you for your comment. We'd like to emphasize that the contributions of our paper go beyond the mere application of Importance Sampling (IS) in DAAs. We aim to provide a **theoretical understanding of overoptimization**, an important problem, in DAAs; based on this understanding, we then propose **Importance Sampling (IS) as an approximation of the KL regularization** between the reference policy and the training policy under the offline setting. In addition, **the roles of IS in PPO (an online algorithm) and in our IS-DPO (an offline algorithm) are also fundamentally different**. Specifically:
>
> - **Foundational Theoretical and Empirical Analysis of IS in DPO**: Our paper provides a theoretical connection of overoptimization and distributional shift between the learned and preference policies in DPO (Section 3.1). Then we propose IS the correction for overoptimization in DPO. However, the most significant analysis of our work is theoreatically demonstrating that IS correction implicitly (and approximately) regularize the DPO's policy in a similar manner as that of minimizing the KL divergence with online samples (see Theorem 1, and Section 5.2's discussion, and also our empirical evidence for online DPO as a response to fRzo's Q2).
> In summary, our paper provides a rigorous theoretical analysis of the overoptimization problem and its solution (via IS) in the context of DAAs, which is the significant contribution beyond simply an application of IS.
>
> - **The roles and impact of Importance sampling and clipping are fundamentally different in PPO and IS-DPO**: While IS has also been used to address distribution shift in PPO, its usage is fundamentally different from our paper. PPO uses clipped importance sampling (IS-clip) to improve sample efficiency. Since each batch is typically discarded after a single gradient update in standard policy gradient methods, IS allows PPO to reuse the same batch multiple times, extracting the most information from each batch. In contrast, our objective is to enforce an effective regularization, to prevent them from pushing up the probability of responses that are out of the offline data distribution, mitigating the over-optimization issue. In addition, in PPO, the clipping mechanism acts as a surrogate trust region to prevent the policy from deviating too far from the previous one (which can help mitigate over-optimization), but in a fully offline setting (as in our paper), this mechanism becomes overly conservative-where gradient is zero when the probability ratio falls outside the clipping range $[1-\epsilon, 1+\epsilon]$, limiting the potential improvement from $\pi_{\text{ref}}$ [4,5]. Our clipping strategy, conversely, mitigates excessively large updates with IS while still providing meaningful updates even when the ratio lies outside the clipping region.
>
> In summary, these contributions are significant and original, and we hope the response has addressed the reviewer's concerns. We thank the reviewer again for the thoughtful feedback.
>
> **Q2: The sensitivity of clipping ratio $\epsilon$ and practical guidance how to set it.**
>
> **Answer:**  Thank you for your question. We'd like to clarify that the ablation studies in Figure 5 already provide practical guidance to set $\epsilon$. Specifically, the studies demonstrate the robustness of IS-DPO's hyperparameter -- specifically that IS-DPO maintains strong performance across a reasonable range of clipping values robustness and in Section 4.1, we empirically suggest the use of $\log \epsilon = 1.0$ as the preferred choice in practice, which we use for all our experiments.
>
> Nevertheless, we follow the reviewer's suggestion and now also provide a more theoretical insight on the relationship between the regularization coefficient $\beta$ (DPO's hyperparameter) and the clipping parameter $\epsilon$. We hope this will provide additional guidance on selecting $\epsilon$.
>
> **Proposition:** *Consider the two distribution $\pi$ an $\pi_{\text{ref}}$. Assuming $\text{Supp}(\pi_\theta)=\text{Supp}(\pi_{\text{ref}})$, then the variance of the importance weight $\mathbb{Var}\left(\frac{\pi_\theta}{\pi_{\text{ref}}}\right)$ under $\pi_{\text{ref}}$ distribution is an upper bound of KL divergence:*
> $$\text{KL}\left(\pi_\theta || \pi_{\text{ref}}\right) \leq \mathbb {Var}\left(\frac{\pi_\theta}{\pi_{\text{ref}}}\right)$$
>
> The proof is straightforward by utilizing the inequality $\log x\leq x-1,\forall x>0$. We will include the detailed proof in our camera-ready version. This proposition highlights that when regularization is weak (i.e., small $\beta$), the policy $\pi_\theta$ can deviate significantly from $\pi_{\text{ref}}$, leading to high variance in the importance weights. This suggests a smaller clipping threshold $\epsilon$ to control variance and stabilize training under such regimes and vice versa when regularization is strong.
>
> We also conduct the relationship between the regularization parameter $\beta \in\\{0.01,0.05,0.1,0.5\\}$ and the clipping ratio $\log\epsilon \in\\{0.25, 0.5, 1.0\\}$. We present the win-rate and KL values below:
>
> **Win-rates**
> | $\beta$ \ $\log\epsilon$ | 0.25 | 0.5|1.0|
> | -------- | -------- | -------- | -------- |
> | 0.01     |  91.2    |  92.0    | 89.7|
> | 0.05     |  92.2    |   93.2  |  94.5 |
> | 0.1    |   93.6 |  94.0  |  96.67  |   |
> |0.5     |  68.9  |   69.7 |   71.0 |    |
>
> **KL values**
> | $\beta$ \ $\log\epsilon$ | 0.25 | 0.5|1.0|
> | -------- | -------- | -------- | -------- |
> | 0.01     |  15.4    |  20.1    | 22.5 |
> | 0.05     |  17.6    |   18.2 |  22.2  |
> | 0.1    |   15.9 |  18.8  | 22.3   |
> |0.5     |  2.55  |   2.78 |   3.06 |    |
>
> **$\epsilon$ as an additional regularization parameter:** We observe that increasing the clipping threshold allows for larger policy updates, which in turn leads to higher KL divergence across all values of $\beta$. Interestingly, at low regularization levels ($\beta=0.01$), using a smaller clipping ratio yields better performance, as it helps prevents over-optimization. In contrast, at higher regularization levels (e.g., $\beta \in\\{0.05, 0.1, 0.5\\}$), larger clipping ratios are more effective. Overall, performance remains robust across a wide range of $\log \epsilon$ values. Consistent with our ablation studies, we recommend setting $\log \epsilon = 1$ as a default choice in practice.
>
>
> **Q3. Comparison with Online-DPO**
>
> **Answer:** Thank you for a great suggestion. As suggested, we conduct experiments to understand how well the IS-DPO can approximate the Online-DPO. We present our findings below:
>
> DPO result:
> |  |  | | | | | |
> | - | - | - | - | - | - | -
> | Win-rate     |   68.8   |    86.9  | 91.4 |86.6| 75.0 |57.6 |
> | $\sqrt{\text{KL}}$ |1.70 | 3.10 | 4.51 | 5.91 | 7.32 | 8.72 |
>
> IS-DPO result:
> |  |  | | | | | |
> | -------- | -------- | -------- | - | - | -| -|
> | Win-rate     |  57.15    |   73.6   | 84.0| 90.0 | 92.8 | 92.78|
> | $\sqrt{\text{KL}}$ |1.46 | 2.37| 3.28| 4.19| 5.10| 6.02 |
>
>
> Online-DPO result:
>
> |  |  | | | | | |
> | -------- | -------- | -------- | -| - | -|- |
> | Win-rate     |  56.11   |  77.06    | 88.7 | 94.07 | 94.7 | 93.6
> | $\sqrt{\text{KL}}$ |1.726 | 3.17| 4.63|6.08 | 7.53 | 8.25
>
> **IS-DPO can closely approximate Online-DPO up to a specific KL divergence threshold:** We observe that IS-DPO also follows a similar performance trajectory to Online-DPO in the low to moderate KL divergence regime $\sqrt{\text{KL}}\left(\pi_\theta ||\pi_{\text{ref}}\right)\approx 6$. In this region, both methods show a similar and steady increase in win rate. This suggests that IS-DPO is capable of closely approximating the behavior and performance of Online-DPO within a certain KL "budget".
>
> **On-policy benefits from large KL "budget"**: While IS-DPO does not show signs of over-optimization, it also does not extrapolate to higher KL regimes, while Online-DPO can find better policies under higher KL regimes. This indicates that on-policy optimization remains beneficial when the optimal policies lie farther from the $\pi_{\text{ref}}$, as Online-DPO is able to leverage a larger KL budget.
>
> **Comparison with DPO:** We also provide a comparison between Online-DPO and DPO. Interestingly, we observe that under low KL budget, DPO appears to outperform Online-DPO $\sqrt{\text{KL}}(\pi_\theta ||\pi_{\text{ref}}) \approx 4.5$. However, it sharply declines as KL increases, indicating over-optimization, while IS-DPO and DPO effectively utilize a larger KL budget to achieve higher performance.
>
> **Q4: Hypothesis on why $\chi$-PO more effect on AlpacaEval.**
>
> **Answer:** Thank you for a great observation. Despite the consistently strong performance of IS-DPO, $\chi$-PO indeed achieves a slightly (though marginally) better result in AlpacaEval. While we do not have a definitive explanation, we suspect IS-DPO's hyperparameter selection (i.e., $\epsilon$) may not be optimal for this dataset. A deeper investigation, e.g., why $\chi$-PO performs well in AlpacaEval, could be valuable, but this requires extensively more experiments and analysis on many benchmarks to confirm this observation. We believe this deserves an independent study beyond the scope of our paper.
>
> ----
>     1. Disentangling Length from Quality in Direct Preference Optimization. ACL24
>     2. Provably Mitigating Overoptimization in RLHF: Your SFT Loss is Implicitly an Adversarial Regularizer. NeurIPS24.
>     3. Correcting the Mythos of KL-Regularization: Direct Alignment without Overoptimization via Chi-Squared Preference Optimization. ICLR25 Spotlight.
>     4. The Sufficiency of Off-Policyness and Soft Clipping: PPO is still Insufficient according to an Off-Policy Measure. AAAI23.
>     5. Off-Policy Proximal Policy Optimization. AAAI23.

---

> > ### Comment · Reviewer_B49u · 2025-08-09
> >
> > Thank the authors for the detailed responses. My major concern on the novelty of the proposed clipping importance-sampling technique is well addressed. I will raise my score accordingly.

---

### Official Review · Reviewer_PDSf · 2025-07-03

**Clarity:** 2
**Significance:** 2
**Originality:** 1
**Rating:** 4
**Confidence:** 4

**Summary:**

This paper addresses the problem of reward over-optimization in offline Direct Alignment Algorithms (DAAs) like DPO. The authors hypothesize that the performance degradation observed in these methods stems from an inadequate implicit KL regularization, which becomes ineffective as the trained policy (π_θ) diverges from the initial reference policy (π_ref). This occurs because the DAA objective is optimized using a static, offline dataset sampled from π_ref, failing to account for the distribution shift during training.

The proposed solution Importance Sampling DAAs (IS-DAAs), directly tackles this issue. It modifies the standard DAA objective by multiplying the loss for each preference pair by an importance sampling ratio. Extensive experiments on summarization and instruction-following tasks demonstrate that IS-DAA significantly mitigates over-optimization, achieves a better performance-vs-KL-divergence trade-off, and outperforms existing regularization methods, especially under low β (KL penalty) settings.

**Questions:**

See weakness.

**Ethical Concerns:**

["NO or VERY MINOR ethics concerns only"]

**Final Justification:**

The theoretical connection between IS correction and implicit KL regularization (Theorem 1) is a good contribution I initially overlooked. So I have raised my score.

**Quality:**

2

**Strengths And Weaknesses:**

Strengths:
- The paper tackles a recognized and important issue in LLM alignment: the tendency of offline Direct Alignment Algorithms (DAAs) like DPO to over-optimize and degrade in performance.
- The proposed method is straightforward, which applies the standard technique of importance sampling to the DPO loss to correct for distribution shift.
- The authors provide empirical results across several benchmarks that show some improvement over baseline DAA methods, particularly in mitigating performance decline during training.

Weakness:
- The core proposed by the authors that offline algorithms suffer from distribution shift as the policy moves away from the data-generating policy, is a well-known problem in the field of offline reinforcement learning. The proposed solution, applying importance sampling, is a standard method to tackle this, which has already applied in PPO. As such, the paper's conceptual contribution feels incremental rather than novel, amounting to the application of a known technique to the specific context of DPO.
- The method's stability and performance rely entirely on the clipping ratio. However, the paper offers no principled way to set this crucial hyper-parameter, aside from a single ablation study shown in Figure 5. This introduces a new, sensitive parameter that a practitioner must optimize via expensive trial-and-error, which counteracts the claimed simplicity of the method and could easily negate its benefits in practice.
- The paper claims the method has "minimal computational overhead," but this is not thoroughly justified. Calculating sequence-level importance ratios for every sample in a batch adds computational steps that, while not requiring new forward passes, are not free. Considering its added complexity, it is not as simple implied by the paper.

---

> ### Author Rebuttal · Authors · 2025-07-31
>
> Thank you for your thoughtful review and valuable feedback. Please find our response to your comments below.
>
> **Q1. The core proposed by the authors that offline algorithms suffer from distribution shift as the policy moves away from the data-generating policy, is a well-known problem in the field of offline reinforcement learning. The proposed solution, applying importance sampling, is a standard method to tackle this, which has already applied in PPO. As such, the paper's conceptual contribution feels incremental rather than novel, amounting to the application of a known technique to the specific context of DPO.**
>
> **Answer:** Thank you for your comment. We'd like to highlight that our contributions goes beyond the use of Importance Sampling (IS) to address the distribution shift, due to the following reasons:
>
> - **Foundational Theoretical and Empirical Analysis of IS in DPO**: Our paper provides a theoretical connection of overoptimization and distributional shift between the learned and preference policies in DPO (Section 3.1). Then we propose IS the correction for overoptimization in DPO. However, the most significant analysis of our work is theoreatically demonstrating that IS correction implicitly (and approximately) regularize the DPO's policy in a similar manner as that of minimizing the KL divergence with online samples (see Theorem 1, and Section 5.2's discussion, and also our empirical evidence for online DPO as a response to fRzo's Q2).
> In summary, our paper provides rigorous theoretical analysis of the overoptimization problem and its solution (via IS) in the context of DAAs, which is the significant contribution beyond simply an application of IS.
>
>
> - **The roles and impact of Importance sampling and clipping are fundamentally different in PPO and IS-DPO**: While IS has also been used to address distribution shift in PPO, its usage is fundamentally different from our paper. PPO uses clipped importance sampling (IS-clip) to improve sample efficiency. Since each batch is typically discarded after a single gradient update in standard policy gradient methods, IS allows PPO to reuse the same batch multiple times, extracting the most information from each batch. In contrast, our objective is to enforce an effective regularization, to prevent the from pushing up the probability of responses that are out of the offline data distribution, mitigating the over-optimization issue. In addition, in PPO, the clipping mechanism acts as a surrogate trust region to prevent the policy from deviating too far from the previous one (which can help mitigate over-optimization), but in a fully offline setting (as in our paper), this mechanism becomes overly conservative-where gradient is zero when the probability ratio falls outside the clipping range $[1-\epsilon, 1+\epsilon]$, limiting the potential improvement from $\pi_{\text{ref}}$ [4,5]. Our clipping strategy, conversely, mitigates excessively large updates with IS while still providing meaningful updates even when the ratio lies outside the clipping region.
>
> In summary, these contributions are significant and original, and we hope the response has addressed the reviewer's concerns. We thank the reviewer again for the thoughtful feedback.
>
> **Q2. The method's stability and performance rely entirely on the clipping ratio. However, the paper offers no principled way to set this crucial hyper-parameter, aside from a single ablation study shown in Figure 5. This introduces a new, sensitive parameter that a practitioner must optimize via expensive trial-and-error, which counteracts the claimed simplicity of the method and could easily negate its benefits in practice.**
>
> **Answers:** Thank you for the comment. IS-DPO indeed has a hyperparameter, much similar to prior works, with multiple hyperparameters, tackling over-optimization or distributional shift [1,2,3]. However, our ablation studies in Figure 5 demonstrate the robustness of IS-DPO's hyperparameter -- specifically that IS-DPO maintains strong performance across a reasonable range of clipping values robustness and in Section 4.1, we empirically suggest the use of $\log \epsilon = 1.0$ as the preferred choice in practice, which we use for all our experiments.
>
> In addition, we will now provide a theoretical insight that better understands the relationship between the regularization coefficient $\beta$ and the clipping parameter $\epsilon$, which helps provide a principled tuning of $\epsilon$.
>
> **Proposition:** *Assuming $\text{Supp}(\pi_\theta)=\text{Supp}(\pi_{\text{ref}})$, then the variance of the importance weight $\mathbb{Var}\left(\frac{\pi_\theta}{\pi_{\text{ref}}}\right)$ under $\pi_{\text{ref}}$ distribution is an upper bound of KL divergence:*
> $$\text{KL}\left(\pi_\theta || \pi_{\text{ref}}\right) \leq \mathbb {Var}\left(\frac{\pi_\theta}{\pi_{\text{ref}}}\right)$$
>
> The proof is straightforward; we will include the detailed proof in our camera-ready version. This proposition highlights that when regularization is weak (i.e., small $\beta$), the policy $\pi_\theta$ can deviate significantly from $\pi_{\text{ref}}$, leading to high variance in the importance weights. This motivates the use of a smaller clipping threshold $\epsilon$ to control variance and stabilize training under such regimes.
>
> Additionally, we also conduct the relationship between the regularization parameter $\beta \in\\{0.01,0.05,0.1,0.5\\}$ and the clipping ratio $\log\epsilon \in\\{0.25, 0.5, 1.0\\}$. We present the win-rate and KL values below:
>
> Win-rate table:
> | $\beta$ \ $\log\epsilon$ | 0.25 | 0.5| 1.0|
> | -------- | -------- | -------- | -------- |
> | 0.01     |  91.2    |  92.0    | 89.7|
> | 0.05     |  92.2    |   93.2  |  94.5 |
> | 0.1    |   93.6 |  94.0  |  91.4  | 91.4   |
> |0.5     |  68.9  |   69.7 |   71.0 |    |
>
> KL table:
> | $\beta$ \ $\log\epsilon$ | 0.25 | 0.5| 1.0 |
> | -------- | -------- | -------- | -------- |
> | 0.01     |  15.4    |  20.1    | 22.5 |
> | 0.05     |  17.6    |   18.2 |  22.2  |
> | 0.1    |   15.9 |  18.8  | 22.3   |
> |0.5     |  2.55  |   2.78 |   3.06 |    |
>
> **$\epsilon$ as an additional regularization parameter:** We observe that increasing the clipping threshold allows for larger policy updates, which in turn leads to higher KL divergence across all values of $\beta$. Interestingly, at low regularization levels (small $\beta$), using a smaller clipping ratio yields better performance, as it helps prevent over-optimization. In contrast, at higher regularization levels (e.g., $\beta = 0.5$), the stronger regularization larger clipping ratios that are more effective.
>
>
> **Q3. The paper claims the method has "minimal computational overhead," but this is not thoroughly justified. Calculating sequence-level importance ratios for every sample in a batch adds computational steps that, while not requiring new forward passes, are not free. Considering its added complexity, it is not as simple as implied by the paper.**
>
> **Answer**: Thank you for your comment. We'd like to emphasize that the computational overhead of calculating the importance ratios is negligible and results in almost the same time complexity as DPO. Despite adding the additional importance ratio calculation step to the original DPO loss, this ratio can be efficiently computed using intermediate quantities already available in the DPO loss calculation. Specifically, as presented in the paper, the importance ratio can be computed in the log-space as follows:
>
> $$w(\mathbf y_w, \mathbf y_l) = e^{\log\pi_\theta (\mathbf y_w)+\log\pi_\theta(\mathbf y_l)-\log\pi_{\text{ref}}(\mathbf y_w)-\log\pi_{\text{ref}}(\mathbf y_l)}$$
>
> Hence, IS-DPO only adds a small calculation, negligible compared to the much more heavier computations in the vanilla DPO. This implementation is also straightforward and simple, requiring only a single additional line of code.
>
> To empirically validate this claim, we report the average training time over 100 steps for both DPO and IS-DPO. We use 4 A100-80GB GPUs, a batch size of 16, and LLaMA-3.2-3B as the base model:
>
> | IS-DPO | DPO |
> | -------- | --------
> | 92.829s $\pm$ 5.144    | 92.571s $\pm$ 4.837
>
> As can be observed, the training time for IS-DPO is nearly identical to that of the vanilla DPO, confirming that our approach is computationally efficient and negligible in overhead.
>
> ----
>     1. Disentangling Length from Quality in Direct Preference Optimization. ACL24
>     2. Provably Mitigating Overoptimization in RLHF: Your SFT Loss is Implicitly an Adversarial Regularizer. NeurIPS24.
>     3. Correcting the Mythos of KL-Regularization: Direct Alignment without Overoptimization via Chi-Squared Preference Optimization. ICLR25 Spotlight.
>     4. The Sufficiency of Off-Policyness and Soft Clipping: PPO is still Insufficient according to an Off-Policy Measure. AAAI23.
>     5. Off-Policy Proximal Policy Optimization. AAAI23.

---

> > ### Comment · Reviewer_PDSf · 2025-08-04
> > **Thanks for the Rebuttal**
> >
> > Thank you for the thorough rebuttal. The theoretical connection between IS correction and implicit KL regularization (Theorem 1) is a good contribution I initially overlooked. The negligible computational overhead (< 0.3% difference) and robust performance across clipping values address my main concerns. I will raise my score.

---

### Official Review · Reviewer_fRzo · 2025-07-08

**Clarity:** 4
**Significance:** 3
**Originality:** 3
**Rating:** 5
**Confidence:** 4

**Summary:**

* The paper proposes to avoid reward hacking in the offline direct reward optimization setting by regularizing through importance sampling.
* Theoretically, KL regularization applied in online reward optimization technique is computationally intensive when extended to the direct reward setting as it requires multiple samples to be generated from the trained policy model. This paper demonstrates that importance sampling using a Clipped product of likelihood ratios to weight the direct reward is sufficient to induce the KL-regularized reward training objective.
* Further, empirical results on popular benchmarks like AlpacaEval and MTBench demonstrate the efficacy of the approach in achieving higher win rates with a lower KL penalty.

**Questions:**

Adding missing evaluations would improve the soundness of the results.

**Ethical Concerns:**

["NO or VERY MINOR ethics concerns only"]

**Final Justification:**

Update after the author response:

The soundness of the results are of value as the paper address the issue of overoptimization / reward hacking in direct preference optimization.

**Limitations:**

Yes

**Quality:**

3

**Strengths And Weaknesses:**

Strengths:

* The theoretical analysis of the regularization proposed is sound with valid comparisons with online KL regularized reward learning
* Empirical results demonstrate comparable or sometime higher win rates at lower KL penalty
* The sample efficiency in training direct reward learning is maintained while regularizing.

Weaknesses:
* The statistical significance of the results in Figure 3 and 4 are not clear as many of the baseline approaches are overlapping, with an exception of a low KL penalty (beta) setting where the importance sampling based approach performs better
* The analysis of the online DPO baseline which is cited as a motivation to approximate is missing in the results. It would be good to understand the headroom in the approximation proposed between this computationally intensive alternative

---

> ### Author Rebuttal · Authors · 2025-07-31
>
> Thank you for your review and positive feedback on our paper. We appreciate your acknowledgment of the clarity and organization of our work.
>
> Please find our response to your comments below.
>
> **Q1. The statistical significance of the results in Figures 3 and 4 are not clear as many of the baseline approaches are overlapping, with the exception of a low KL penalty (beta) setting where the importance sampling-based approach performs better.**
>
> **Answer:** Thank you for your comment. We will revise the figures in the camera-ready version to clearly indicate statistical improvements. We'd also like to clarify that our evaluation setup follows prior works [Rafailov et al. 2024, Tang et al. 2024a], where Figures 3/4 illustrate the trade-off between KL divergence and the policy performance for all the methods. We kindly ask the reviewer to disregard the visual overlap in the plots. The key takeaway from these figures is that "the other approaches suffer from an early convergence problem, while IS-DPO continues to improve or stay flat at later epochs with a significantly lower KL budget." (lines 241-243). Consequently, the final win-rate results (summarized below) demonstrate the effectiveness of IS-DPO:
>
> | Method | $\beta=0.01$| $\beta=0.05$ | $\beta=0.1$|
> | -------- | -------- | -------- | - |
> |    IS-DPO  | **91.57**   | 93.35     | **94.8**
> | Length-DPO| 60.25 | 88.67 | 88.83|
> | RPO | 62.37  | 88.25 | 88.22
> | DPO | 62.98 | 89.65 | 91.73
> |$\chi$-PO | 76.46 | **93.75** | 93.06
>
>
> **Q2. The analysis of the online DPO baseline, which is cited as a motivation to approximate, is missing in the results. It would be good to understand the headroom in the approximation proposed between this computationally intensive alternative. Adding missing evaluations would improve the soundness of the results.**
>
> **Answer:** Thank you for a great suggestion. We have conducted experiments to understand how well the IS-DPO can better approximate the Online-DPO. As we're not allowed to include a Figure, we provide the (reduced) tabular version of the results instead and will add the scaling law curve version of Online-DPO in our camera-ready version. Our findings are below:
>
> **IS-DPO can closely approximate Online-DPO up to a specific KL divergence threshold:** We observe that IS-DPO also follows a similar performance trajectory to Online-DPO in the low to moderate KL divergence regime $\sqrt{\text{KL}}\left(\pi_\theta ||\pi_{\text{ref}}\right)\approx 6$. In this region, both methods show a similar and steady increase in win rate. This suggests that IS-DPO is capable of closely approximating the behavior and performance of Online-DPO within a certain KL "budget".
>
> **On-policy benefits from large KL "budget"**: While IS-DPO does not show signs of over-optimization, it also does not extrapolate to higher KL regimes, while Online-DPO can find better policies under higher KL regimes. This indicates that on-policy optimization remains beneficial when the optimal policies lie farther from the $\pi_{\text{ref}}$, as Online-DPO is able to leverage a larger KL budget.
>
> **Comparison with DPO:** We also provide a comparison between Online-DPO and DPO. Interestingly, we observe that under a low KL budget, DPO appears to outperform Online-DPO $\sqrt{\text{KL}}(\pi_\theta ||\pi_{\text{ref}}) \approx 4.5$. However, it sharply declines as KL increases, indicating over-optimization, while IS-DPO and DPO effectively utilize a larger KL budget to achieve higher performance.
>
> DPO result:
> |  |  | | | | | |
> | - | - | - | - | - | - | -
> | Win-rate     |   68.8   |    86.9  | 91.4 |86.6| 75.0 |57.6 |
> | $\sqrt{\text{KL}}$ |1.70 | 3.10 | 4.51 | 5.91 | 7.32 | 8.72 |
>
> IS-DPO result:
> |  |  | | | | | |
> | -------- | -------- | -------- | - | - | -| -|
> | Win-rate     |  57.15    |   73.6   | 84.0| 90.0 | 92.8 | 92.78|
> | $\sqrt{\text{KL}}$ |1.46 | 2.37| 3.28| 4.19| 5.10| 6.02 |
>
>
> Online-DPO result:
>
> |  |  | | | | | |
> | -------- | -------- | -------- | -| - | -|- |
> | Win-rate     |  56.11   |  77.06    | 88.7 | 94.07 | 94.7 | 93.6
> | $\sqrt{\text{KL}}$ |1.726 | 3.17| 4.63|6.08 | 7.53 | 8.25

---

> > ### Comment · Reviewer_fRzo · 2025-08-08
> >
> > Thanks for addressing the concerns raised. I look forward to them in the revised version.
> >
> > In the table presented above, could you report confidence intervals or mean +/- standard error by re-running experiments for a fixed \beta? That would further improve the soundness of the results presented.

---

> > > ### Author Response · Authors · 2025-08-09
> > > **Report confidence intervals or standard error**
> > >
> > > Thank you very much for your reply. Please find our response to your comments below.
> > >
> > > **Q1. In the table presented above, could you report confidence intervals or mean +/- standard error by re-running experiments for a fixed $\beta$ ? That would further improve the soundness of the results presented.**
> > >
> > > **Answer:** Thank you for your suggestion. We've also added standard error on 3 different random seeds, with the following results:
> > >
> > > | Method | $\beta=0.01$| $\beta=0.05$ | $\beta=0.1$|
> > > | -------- | -------- | -------- | - |
> > > |    IS-DPO  | **91.57 $\pm$ 1.17**   | 93.35 $\pm$ 0.96     | **94. 8 $\pm$ 0.320**
> > > | Length-DPO| 60.25 $\pm$ 0.73 | 88.67 $\pm$ 0.39 | 88.83 $\pm$ 0.369|
> > > | RPO | 62.37 $\pm$ 2.25  | 88.25 $\pm$ 1.32 | 88.22 $\pm$ 0.20
> > > | DPO | 62.98 $\pm$ 2.18 | 89.65 $\pm$ 0.29 | 91.73 $\pm$ 0.394
> > > |$\chi$-PO | 76.46 $\pm$ 1.77 | **93.75 $\pm$ 0.51** | 93.06 $\pm$ 0.322

---

### Note · Authors · 2025-08-16

We sincerely thank all reviewers for their constructive feedback. We'd like to summarize our discussions as follows:

**(1) The novelty of Importance sampling in IS-DPO and PPO:** Rev. PDSf and B49u inquired about the novelty of our importance sampling method. We highlighted our unique contributions: we provided a theoretical foundation of the use of importance sampling to address the overoptimization problem via Theorem 1, showing IS-DPO can *approximately regularize explicit KL regularization* as in standard RLHF. We also highlighted its fundamentally different role in IS-DPO and PPO, where PPO uses clipped importance sampling to improve sample efficiency and cannot be applied in an offline setting due to conservatism, while IS-DPO aims to enforce a more efficient KL regularization and can provide meaningful updates in the offline setting.

**(2) The impact of the clipping hyper-parameter $\epsilon$ and computational overhead:** Rev. PDSf and B49u also raised questions about the sesitiveness of an additional hyper-parameter $\epsilon$ and it computational time. We show that the computational time of calculating the importance weight is negligible, resulting in a nearly identical complexity of IS-DPO to DPO. Furthermore, we also provided an additional analysis for choosing the clipping threshold $\epsilon$, where a small $\epsilon$ helps prevent over-optimization while a bigger $\epsilon$ allows for larger updates.

**(3) The Taylor Expansion Analysis:** Rev. VYu7 and PaV2 questioned about the Taylor approximation around $0$, which is not valid when the learned policy deviates away from $\pi_{\text{ref}}$. We explained that the Taylor approximation aims to show that DAAs can only enforce KL regularization when the learned policy is close to $\pi_{\text{ref}}$ and cannot prevent over-optimization when it diverges further without importance sampling.

**(4) Technical Details:** we also responded to other comments from Rev. VYu7, fRzo, and PaV2 and:
- Provided an additional analysis with Online-DPO objective.
- Discussed and added experiments around the assumption that the offline preference data is generated by the reference policy.
- Expanded Section 4.3, where IS-DPO serves as an additional support constraint to avoid out-of-support of $\pi_{\text{ref}}$ extrapolation.
- Added additional discussions about principles of pessimism.

We're again grateful for your comments and hope to receive your positive support for the paper in the final evaluation.

---

### Decision · Program_Chairs · 2025-09-17

**Decision:**

Accept (poster)

**Comment:**

This paper considers the problem of reward over-optimization in offline direct alignment algorithms like DPO. The main idea behind their approach is to take standard DAAs like DPO and multiple the loss by a clipped importance weight to address distribution shift between the trained policy \pi_theta and \pi_ref. The authors provide promising empirical results for this approach on standard RLHF datasets, and show via some simple calculations that the method can be interpreted as implicitly enforcing KL regularization.

Reviewers initially were mixed, with many additional questions around experiments and derivation. However, after the discussion period, they were uniformly fairly positive and recommended acceptance. I strongly encourage the authors to include all of the additional derivations and experiments that came up in discussion.